# Analysis of the SNARE Stx8 recycling reveals that the retromer-sorting motif has undergone evolutionary divergence

Francisco Yanguas[1,2], M.-Henar Valdivieso[1,2]*

**1** Departamento de Microbiología y Genética, Universidad de Salamanca. Salamanca. Spain, **2** Instituto de Biología Funcional y Genómica (IBFG), Consejo Superior de Investigaciones Científicas (CSIC). Salamanca. Spain

\* henar@usal.es

## Abstract

Fsv1/Stx8 is a *Schizosaccharomyces pombe* protein similar to mammalian syntaxin 8. *stx8Δ* cells are sensitive to salts, and the prevacuolar endosome (PVE) is altered in *stx8Δ* cells. These defects depend on the SNARE domain, data that confirm the conserved function of syntaxin8 and Stx8 in vesicle fusion at the PVE. Stx8 localizes at the *trans*-Golgi network (TGN) and the prevacuolar endosome (PVE), and its recycling depends on the retromer component Vps35, and on the sorting nexins Vps5, Vps17, and Snx3. Several experimental approaches demonstrate that Stx8 is a cargo of the Snx3-retromer. Using extensive truncation and alanine scanning mutagenesis, we identified the Stx8 sorting signal. This signal is an IEMeaM sequence that is located in an unstructured protein region, must be distant from the transmembrane (TM) helix, and where the $^{133}$I, $^{134}$E, $^{135}$M, and $^{138}$M residues are all essential for recycling. This sorting motif is different from those described for most retromer cargoes, which include aromatic residues, and resembles the sorting motif of mammalian polycystin-2 (PC2). Comparison of Stx8 and PC2 motifs leads to an IEMxx(I/M) consensus. Computer-assisted screening for this and for a loose Ψ(E/D)ΨXXΨ motif (where Ψ is a hydrophobic residue with large aliphatic chain) shows that syntaxin 8 and PC2 homologues from other organisms bear variation of this motif. The phylogeny of the Stx8 sorting motifs from the *Schizosaccharomyces* species shows that their divergence is similar to that of the genus, showing that they have undergone evolutionary divergence. A preliminary analysis of the motifs in syntaxin 8 and PC2 sequences from various organisms suggests that they might have also undergone evolutionary divergence, what suggests that the presence of almost-identical motifs in Stx8 and PC2 might be a case of convergent evolution.

## Author summary

Eukaryotes possess membranous intracellular compartments, whose communication is essential for cellular homeostasis. Protein complexes that facilitate the generation, transport, and fusion of coated vesicles mediate this communication. Since alterations in these

---

**Data Availability Statement:** All relevant data are within the manuscript and its Supporting Information files.

**Funding:** MHV received a grant from Ministerio de Economía, Industria y Competitividad (Spain. https://www.mineco.gob.es/)/European Regional Development Fund (FEDER) program. Grant BFU2017-84508-P. IBFG (Instituto de Biología Funcional y Genómica) received a grant from the Junta de Castilla y Leon (https://www.educa.jcyl. es/universidad/es)/ FEDER program Escalera de Excelencia Grant CLU-2017-03/14-20. FY was supported by an FPU fellowship from the Spanish Ministry of Education. The funders had no role in study design, data collection and analysis, decision to publish, or preparation of the manuscript.

**Competing interests:** The authors have declared that no competing interests exist

processes lead to human disease, their characterization is of biological and medical interest. Retromer is a protein complex that facilitates retrograde trafficking from the prevacuolar endosome to the Golgi, being essential for the functionality of the endolysosomal system. SNAREs are required for vesicle fusion and, after facilitating membrane merging, are supposed to return to their donor organelle for new rounds of fusion. However, little is known about this recycling. We have found that Stx8, a fungal SNARE similar to human syntaxin 8, is a retromer cargo, and have identified its retromer binding motif. Sequence screening and comparison has determined that this sorting motif is conserved mainly in fungal Stx8 sequences. Notably, this motif is similar to the retromer sorting motif that is present in a family of vertebrate ion transporters. Our initial phylogenetic analyses suggest that, although retromer and some of its cargoes are conserved, the sorting motif in the cargoes might have undergone evolutionary divergence.

## Introduction

Protein trafficking between the late Golgi and lysosomes (vacuoles in plants and fungi) is essential for cellular homeostasis. Several protein complexes, including vesicle coats and adaptors, tethers, and soluble N-ethyl maleimide sensitive factor attachment protein receptors (SNAREs) participate in this traffic [1,2]. SNAREs are central components of the fusion machinery that facilitate the merging of vesicle and organelle membranes [3]. They are small, generally tail-anchored proteins defined by the presence of a SNARE domain (a conserved 60–70 amino acid α-helix with heptad repeats that can form coiled-coil structures) proximal to their C-terminus. Although initially classified as v- or t-SNAREs (present in vesicles or target membranes), they are now termed R- and Q-SNAREs depending on the presence of an arginine or a glutamine at the central position of the conserved motif. Q-SNAREs are subdivided into Qa, Qb, and Qc according to the sequence similarity of their SNARE domain. Membrane fusion requires the formation of a stable *trans*-SNARE through the interaction between one R-, one Qa-, one Qb-, and one Qc-SNARE [4,5]. The structure of the SNARE N-terminal region is more variable; it sometimes includes additional coiled-coil domains that are not directly involved in membrane fusion. Specific SNARE interactions facilitate vesicle fusion at different organelles [5,6]. According to analyses of their subcellular localization and physical interactions, the syntaxin 7 (Stx7, Qa)-Vti1b (Qb)-syntaxin 8 (Stx8. Qc)-Vamp8 (R) complex promotes homotypic fusion of late endosomes, and Stx7-Vti1b-Stx8-Vamp7 promotes the fusion of late endosomes to vacuoles, as well as homotypic vacuole fusion [4,6–8]. Although SNAREs are expected to recycle back to the donor organelle for new rounds of vesicle fusion after they disassemble [1,5], there is little information about this recycling.

Retromer has been described in yeast as a vesicle coat that mediates retrograde trafficking from the late, prevacuolar endosome (PVE), to the *trans*-Golgi network (TGN, the yeast equivalent of early endosomes; [9,10]) and to plasma membrane. It was later shown to be conserved in mammals [11,12]. Retromer is composed of a Vps26-Vps29-Vps35 core, which is considered to be the cargo-selective complex (CSC; [12–17]) and two sorting nexins (SNXs) that interact with phospholipids, deform membranes, and mediate retromer recruitment to membranes [18,19]. The first identified retromer-associated SNXs were ScVps5 (SNX1 or SNX2 in mammals) and ScVps17 (SNX5 or SNX6). Later research has reported other SNXs that participate in protein recycling from the PVE and contribute to cargo selection; some of these SNXs associate with the CSC to generate alternative retromer complexes [20–29]. The retromer cargoes that were first identified were mammalian cation-independent mannose 6-phosphate

receptor (CI-MPR; [12,13, 16,17]) and its yeast counterpart Vps10. Subsequently, other yeast proteins have been shown to interact with retromer components and/or to depend on retromer for their recycling. Regarding SNAREs, Pep12 is not recycled to the TGN in the absence of Snx3-retromer; and retromer, Snx4, and Snx42 contribute to the recycling of Snc1 to plasma membrane [30–32]. Nevertheless, their physical interaction with retromer has not been shown, nor has their sorting motif been identified. Several mammalian proteins have also been identified as retromer cargoes, and aberrant retromer function has been related to alterations in development and to neurological diseases and cancer [33]. Therefore, studying the requirements for retromer interaction with cargoes is a subject of biological and medical interest.

In this work, we have used the model system *Schizosaccharomyces pombe* to perform a detailed study of the function and regulation of the SNARE Fsv1/Stx8, homologue of syntaxin 8. Fsv1 was identified as a SNARE, required for optimal Cpy1 sorting, that collaborates with the VAM7 homologue Vsl1 in TGN to vacuole traffic [34,35]. It shares 18% identity and 41% similarity with human syntaxin 8, and Fsv1 SNARE domain is 29% identical and 53% similar to that in syntaxin 8. Due to this similarity, we will refer to this gene as *stx8*[+] throughout the text. Our results show that this SNARE is required for PVE integrity and for optimal growth under saline stress. Stx8 is localized at the TGN and the PVE, and its retrograde trafficking depends on the retromer CSC and on Vps5, Vps17, and Snx3. We have narrowed its Snx3-retromer sorting motif to a GsdIEMeaM sequence that is different from the motifs found in other retromer cargoes, which normally include bulky aromatic residues [26,29,36–39], but similar to the GxxIEMQxIx sorting motif found in polycystin-2 homologues [40]. Screening Snx3-retromer potential cargoes for the presence of this and some related sorting motifs indicated that the signals recognized by retromer might have undergone evolutionary divergence. Moreover, *Saccharomyces cerevisiae* Pep12, a known Snx3-retromer cargo, lacks this and other retromer sorting motifs described. Thus, our results expand the repertoire of retromer sorting motifs and show that this repertoire is still incomplete.

## Results

### Fsv1/Stx8 is required for correct morphology and functionality of the PVE

Several experiments have been performed to obtain more information about the function of Fsv1/Stx8. Given that Cpy1 is partially missorted in the *fsv1Δ* mutant [35], we investigated the distribution of Vps10-green fluorescent protein (GFP). In the wild type (WT) strain, this receptor was distributed in several intensely bright cytoplasmic dots that mostly correspond to the PVE [41]. In the *stx8Δ* strain, the fluorescence intensity of the dots was considerably reduced (Fig 1A), while the protein level was similar to that in the WT (Fig 1B). Nevertheless, the fact that Cpy1 processing was only partially defective in a *fsv1Δ* mutant [35] and that we observed this carboxypeptidase in the vacuoles of *stx8/fsv1Δ* cells (Fig 1C) indicated that the Vps10-Cpy1 complex at least partially reached the PVE-vacuole interface in the absence of this SNARE. Given that Vps10 recycling depends on the retromer [42], we analyzed the localization of Vps10 in a retromer mutant lacking Stx8 to confirm this issue. As shown in Fig 1D, we observed Vps10-GFP at the vacuole surface in the *vps35Δ* and *stx8Δ vps35Δ* strains, demonstrating that in the absence of the SNARE the receptor was delivered to the PVE, from where it was recycled to the TGN by retromer. The level of Vps10-GFP in *vps35Δ* was similar to that in the WT (Fig 1B), and the levels of Stx8 and Vps35 proteins were not altered in the *vps35Δ* and stx8Δ strains, respectively (S1 Fig). The fact that Vps10 was also observed at the vacuole surface in the *stx8Δ vps35Δ apm3Δ* strain—lacking Stx8, retromer, and the μ subunit of the AP-3 adaptor [43]—(Fig 1D) showed that it was not delivered from the TGN to the vacuole along the alkaline phosphatase (ALP) pathway bypassing the PVE [2]. Additionally, to investigate

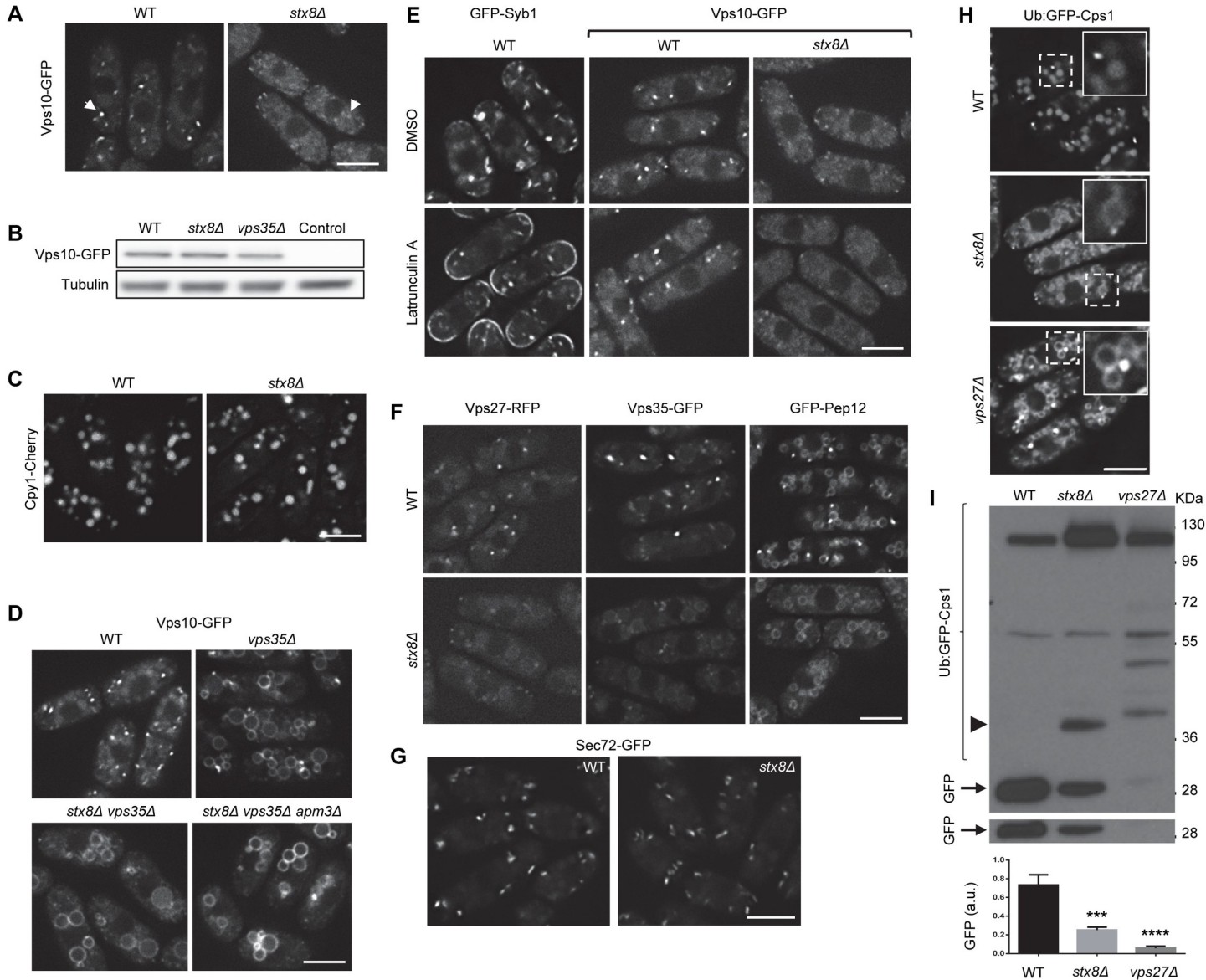

**Fig 1. Stx8 is required for the morphology and functionality of the prevacuolar endosome (PVE).** (A) Localization of Vps10-GFP in wild type (WT) and *stx8Δ* cells. Arrowheads denote cytoplasmic fluorescent dots. (B) The level of Vps10-GFP in cell extracts from the WT, *stx8Δ* and *vps35Δ* strains was analyzed by western blot. Cell extracts from a strain that did not express Vps10-GFP was used as a negative control. Tubulin was used as loading control. (C) Cpy1-Cherry distribution in the same strains as in (A). (D) Vps10-GFP distribution in the indicated strains. (E) WT cells bearing either GFP-Syb1 or Vps10-GFP, and *stx8Δ* cells bearing Vps10-GFP were treated with dimethyl sulfoxide (DMSO; solvent) and with latrunculin A for 20 minutes and photographed. (F) Distribution of the indicated PVE markers in WT and *stx8Δ* cells. (G) Distribution of the *trans*-Golgi network (TGN) marker Sec72-GFP in WT and *stx8Δ* cells. (H) Distribution of Ub:GFP-Cps1 in the indicated strains. For each image, the inset in the upper-right corner corresponds to an enlargement of the region delimited by dashed lines. (I) Equal amounts of protein in cell extracts from the indicated strains bearing Ub:GFP-Cps1 were subjected to western blotting with anti-GFP. The arrowhead denotes a 40 kDa band present in *stx8Δ*, and the arrow denotes the clipped GFP. The lower panel shows a low exposure of the gel region where the clipped GFP was detected. The experiment was performed three times. The lowest panel shows the mean, standard deviation, and statistical significance of the differences in the amount of GFP (a.u., arbitrary units), determined by Dunnett's multiple comparisons tests after analysis of variance (ANOVA). ***, $p < 0.001$; ****, $p < 0.0001$. Images are single planes captured with a DeltaVision system. Scale bar, 5 μm.

whether Vps10 was diverted to the plasma membrane, from where it would be endocytosed, we treated the WT and *stx8Δ* strains with latrunculin A. We analyzed the distribution of GFP-Syb1 (the *S. pombe* synaptobrevin/Snc1 homologue) as a control for the treatment. As shown in Fig 1E, we observed GFP-Syb1 at the cell surface of the WT cells in the presence of

the drug while we did not observe Vps10-GFP at the cell surface of either the untreated WT cells or the *stx8Δ* cells treated or not with latrunculin A. Taken together, these results demonstrated that, although there were no intensely bright fluorescent dots in the cytoplasm, Vps10-GFP cycled between the TGN and the PVE in the absence of Stx8.

To distinguish between the possibilities that Vps10 did not accumulate in the PVE or that this organelle was defective, we analyzed the distribution of several PVE markers in *stx8Δ* cells. In the WT cells, there were prominent cytoplasmic dots bearing the endosomal sorting complexes required for transport (ESCRT) component Vps27 and the retromer subunit Vps35, and the SNARE Pep12 appeared at dots and at the vacuole membrane (Fig 1F). In the *stx8Δ* mutant, the distribution of all these proteins was altered such that we did not observe prominent cytoplasmic dots in any case. These results showed that the composition/morphology/ organization of the PVE was abnormal in the absence of Stx8. On the other hand, the vacuolar Pep12 distribution was not altered, and the vacuoles appeared normal, in agreement with previous results [35]. Retromer and its cargoes traffic to the TGN; consequently, we assessed the morphology of this organelle in WT and *stx8Δ* cells by analyzing the distribution of the Arf1 GEF Sec72-GFP. As shown in Fig 1G, there were no apparent differences in the distribution of this TGN marker between both strains. These results strongly suggested that the absence of Stx8 altered the PVE but not the other organelles that communicate with it. To examine the functionality of the PVE, we used the Ub:GFP-Cps1 construct [41] to analyze the distribution and processing of Cps1, a vacuolar carboxy-peptidase delivered from the PVE to the vacuole in a ESCRT-dependent manner [44,45]. We observed this protein in bright cytoplasmic dots and inside the vacuoles in WT cells, as well as in prominent cytoplasmic dots and the vacuolar surface in *vps27Δ* cells (Fig 1H; [41]). In the *stx8Δ* mutant, the Cps1 distribution was different. There were low-intensity fluorescence dots in the cytoplasm, like those observed for Vps10-GFP, Vps27-RFP, and Vps35-GFP (Fig 1A and 1F). Additionally, the vacuole surface and interior were decorated with Ub:GFP-Cps1 (Fig 1H). Regarding its processing, we used western blotting to evaluate the accumulation of free GFP, which is produced by Ub:GFP-Cps1 proteolytic cleavage by the Isp6 and Psp3 vacuolar proteases (S2 Fig). As shown in Fig 1I, this accumulation was significantly lower in the *stx8Δ* strain compared with the WT; nevertheless, the processing defect was not as strong as in *vps27Δ*. Additionally, the *stx8Δ* strain presented a degradation band different from those observed in the *vps27Δ* strain (arrowhead in Fig 1I). These results showed a partial defect in Cps1 processing. The defects in Cpy1 maturation [35] and in Cps1 processing (Fig 1H and 1I) showed that the PVE functionality was defective in the absence of Stx8. Nevertheless, this defect was partial, suggesting that other SNARE might substitute for Stx8. Vsl1 is a Qc-SNARE that collaborates with Stx8 in trafficking from the TGN to the vacuole [34]. However, the fact that there were no apparent differences in the distribution of Vps10, Pep12, and Ub:GFP-Cps1 in WT and *vsl1Δ* cells—and that *vsl1*[+] overexpression did not correct the defect in Vps10-GFP distribution in *stx8Δ* cells (S3 Fig)—showed that the VAM7 homologue Vsl1 was not partially redundant with Stx8 in these processes.

## The SNARE function of Stx8 is required for PVE integrity and growth under saline stress

The endomembrane system plays an important role in response to amino acid starvation and in maintaining osmotic, pH, and ionic homeostasis [46–48]. To evaluate whether the defect in the PVE detected by microscopy and western blotting was relevant for the physiology of *stx8Δ* cells, we compared the growth of the *stx8Δ* strain under different stress conditions to that of the WT, the retromer mutant *vps35Δ*, and the ESCRT mutant *vps27Δ* strains. As shown in Fig 2A, the *vps35Δ* strain exhibited reduced growth on minimal medium with phenylalanine

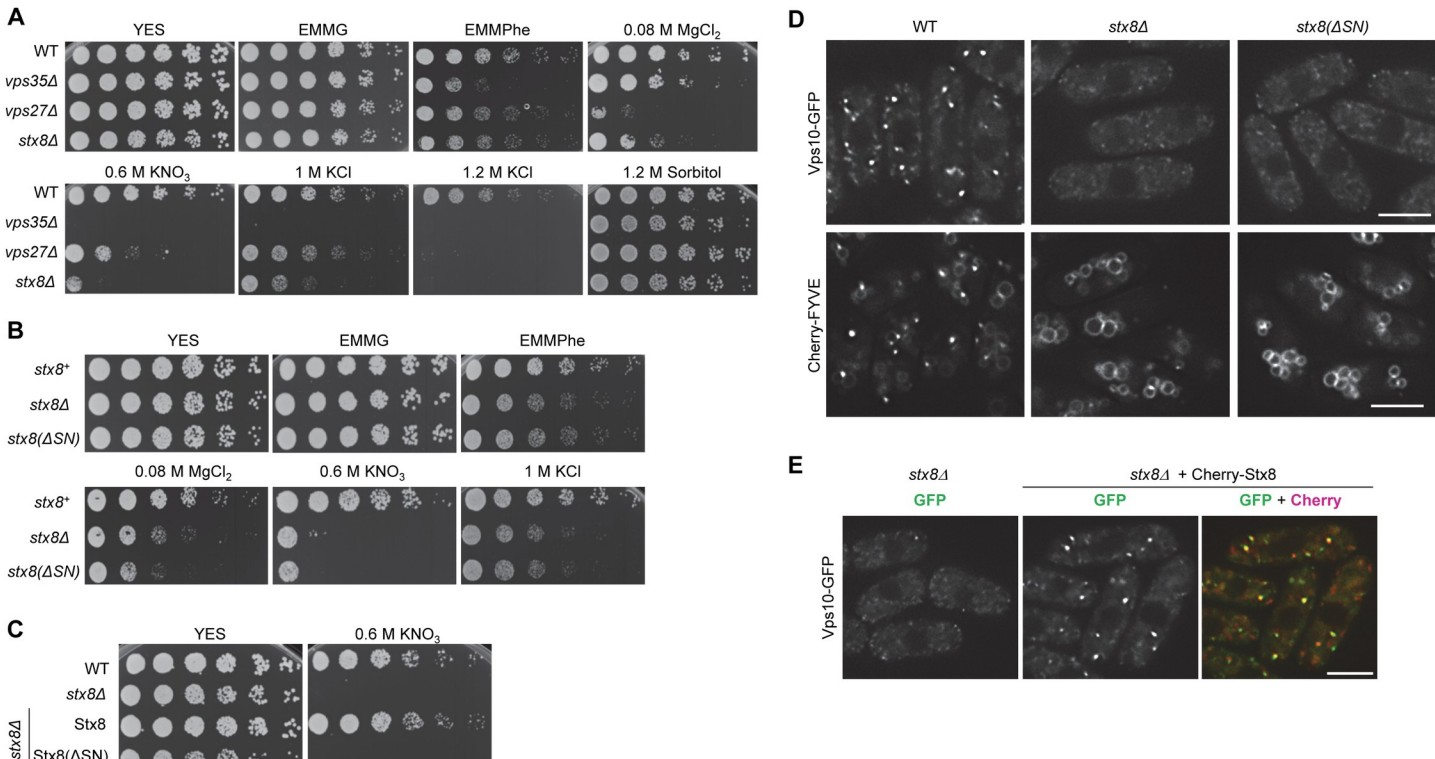

**Fig 2. The SNARE domain is essential for Stx8 function.** (A) $3 \times 10^4$ cells and serial 1:4 dilutions from the indicated strains were spotted on YES (rich medium), EMMG (minimal medium with a rich nitrogen source), or EMMPhe (minimal medium with a poor nitrogen source) plates; YES plates were supplemented with the indicated compounds. Plates were incubated at 32°C for 3 days. (B) The same experimental details as in A, but the strains were *stx8+* (wild type [WT] control), *stx8Δ* (null mutant), and *stx8(ΔSN)* (a *stx8* mutant lacking the SNARE domain). (C) The same experimental details as in A, but the strains were WT, *stx8Δ*, and *stx8Δ* bearing integrated plasmids for the expression of either GFP-Stx8 or GFP-Stx8(ΔSN). (D) *stx8+*, *stx8Δ*, and *stx8(ΔSN)* cells bearing Vps10-GFP (upper panels) and Cherry-FYVE (lower panels) were photographed with a DeltaVision system. (E) Images of *stx8Δ* cells expressing either Vps10-GFP alone or Vps10-GFP and Cherry-Stx8 acquired with a DeltaVision System. In (D) and (E) images are single planes. Scale bar, 5 μm.

(EMMPhe, a poor nitrogen source; [49,50]) while the growth defect of *vps27Δ* and *stx8Δ* strains in this medium was marginal. Regarding growth on plates with salts, the *vps35Δ* strain exhibited a mild sensitivity to $MgCl_2$ and a strong sensitivity to potassium salts; the ESCRT mutant was very sensitive to magnesium and slightly sensitive to potassium. The *stx8Δ* strain exhibited sensitivity to potassium and magnesium salts. Finally, none of the mutants was sensitive to sorbitol, an osmotically active compound. These results showed that Stx8 was required for an adequate response to ionic stress.

Stx8 was required for the morphology and functionality of the PVE (Fig 1A and 1F), as well as for growth in the presence of saline stress (Fig 2A). To confirm that these phenotypes were the consequence of the loss of Stx8 SNARE function, we integrated a *stx8(ΔSN)* variant that lacked the SNARE domain into the *stx8+* locus under the control of its endogenous promoter. We then compared the growth of the *stx8(ΔSN)* mutant under stress conditions to that of the WT and the *stx8Δ* null mutant strains. As shown in Fig 2B, the growth of *stx8(ΔSN)* and *stx8Δ* mutants on EMMPhe plates, and in the presence of $MgCl_2$, $KNO_3$, and KCl, was similar. Furthermore, reintroducing Stx8 but not Stx8(ΔSN) reverted potassium sensitivity to the *stx8Δ* mutant (Fig 2C), which confirmed that the lack of Stx8 SNARE function produced this phenotype. Next, we compared the distribution of Vps10-GFP and the PVE marker Cherry-FYVE in the *stx8(ΔSN)* mutant to that in the WT and *stx8Δ* cells. As shown in Fig 2D, for both markers the fluorescence distribution in the *stx8(ΔSN)* strain was similar to that in the *stx8Δ* strain and

different from that in the WT strain. The expression a Cherry-Stx8 fusion protein in the *stx8Δ* strain that bears Vps10-GFP reverted the defect in the receptor localization (Fig 2E). These results showed that the defects in the morphology and functionality of the PVE in *stx8Δ* cells were a consequence of the loss of Stx8 function in vesicle fusion at the PVE. Additionally, colocalization between Cherry-Stx8 and Vps10-GFP (Fig 2E) suggested that the SNARE might accumulate in the PVE.

## Retromer and Snx3 are required for Stx8 recycling from the PVE to the TGN

To gain information about the regulation of Stx8, we determined its subcellular distribution. Analysis of GFP-Stx8 colocalization with the TGN and PVE markers Cfr1 and Vps27, respectively [41,51,52], showed that this SNARE localized in both organelles, with accumulation in the PVE (Fig 3A). This finding agrees with its colocalization with Vps10 and with results obtained for syntaxin 8, which is more abundant in endosomal than Golgi membranes [53,54]. We determined whether Stx8 cycled between the TGN and PVE through delivery to the plasma membrane and posterior endocytosis. To do so, we analyzed its localization in the presence of latrunculin A. As shown in Fig 3B, this treatment led to the accumulation of Syb1 but not Stx8 at the cell surface, data that ruled out our hypothesis. Next, we analyzed the possibility that retromer recycled this SNARE by observing Stx8 distribution in *vps35Δ*. In this mutant, we observed Stx8 in cytoplasmic dots and in multiple circular structures of various sizes. To confirm that these structures were vacuoles, we incubated cells in the presence of FM4-64. Observation under the microscope and line-scan analysis demonstrated that GFP-Stx8 and FM4-64 colocalized in *vps35Δ* but not in WT cells (Fig 3C). In agreement, the colocalization between GFP-Stx8 and Cfr1-RFP was significantly reduced in *vps35Δ* (Fig 3D). These results showed that the retromer CSC was required to recycle the SNARE from the PVE to the TGN and indicated that Stx8 might be a retromer cargo. To further explore this possibility, we used different approaches; first, we analyzed colocalization between Cherry-Stx8 and Vps35-GFP. The results showed that Stx8 and Vps35 colocalized (Fig 3E). Next, we analyzed physical interaction by coimmunoprecipitation. The strains expressing GFP-Stx8 bore a *stx8Δ* deletion to avoid competition between tagged and untagged proteins. The localization of the SNARE with this background was the same as with the *stx8+* background. Additionally, we had deleted *vps28+* in these strains. In ESCRT mutants, there is a defect in protein sorting at the PVE [44,55]; therefore, it is expected that the contact between proteins in this organelle last longer in *vps28Δ* than in WT cells [56]. This phenomenon would facilitate detection of the Stx8-Vps35 complex. Under these conditions, there was physical interaction between GFP-Stx8 and Vps35-HA (Fig 3F). Finally, we determined direct interaction by two-hybrid analysis using a Stx8 variant that lacked the transmembrane helix, and that we termed Stx8*. We could not detect interaction between Stx8 and Vps26 (S4 Fig). Nevertheless, we could detect direct interaction between Stx8* and the retromer CSC using a fusion protein composed of Vps29, Vps35, and Vps26 (Fig 3G. See Materials and methods). Together, these results showed that Stx8 is a retromer cargo.

## Requirements for Stx8 recycling by retromer

To gain information about the requirements for Stx8 recycling from the PVE, we determined the sorting nexins implicated in the process by analyzing Stx8 distribution in *vps5Δ*, *vps17Δ*, *snx3Δ*, and *snx4Δ* mutants. The SNARE accumulated at the vacuolar surface in *vps5Δ*, *vps17Δ*, and *snx3Δ* cells, but not in *snx4Δ* cells (Fig 4A). In agreement, Stx8 coimmunoprecipitation with Vps35 was reduced in *vps5Δ* and *snx3Δ* mutants (Fig 4B), showing that both nexins

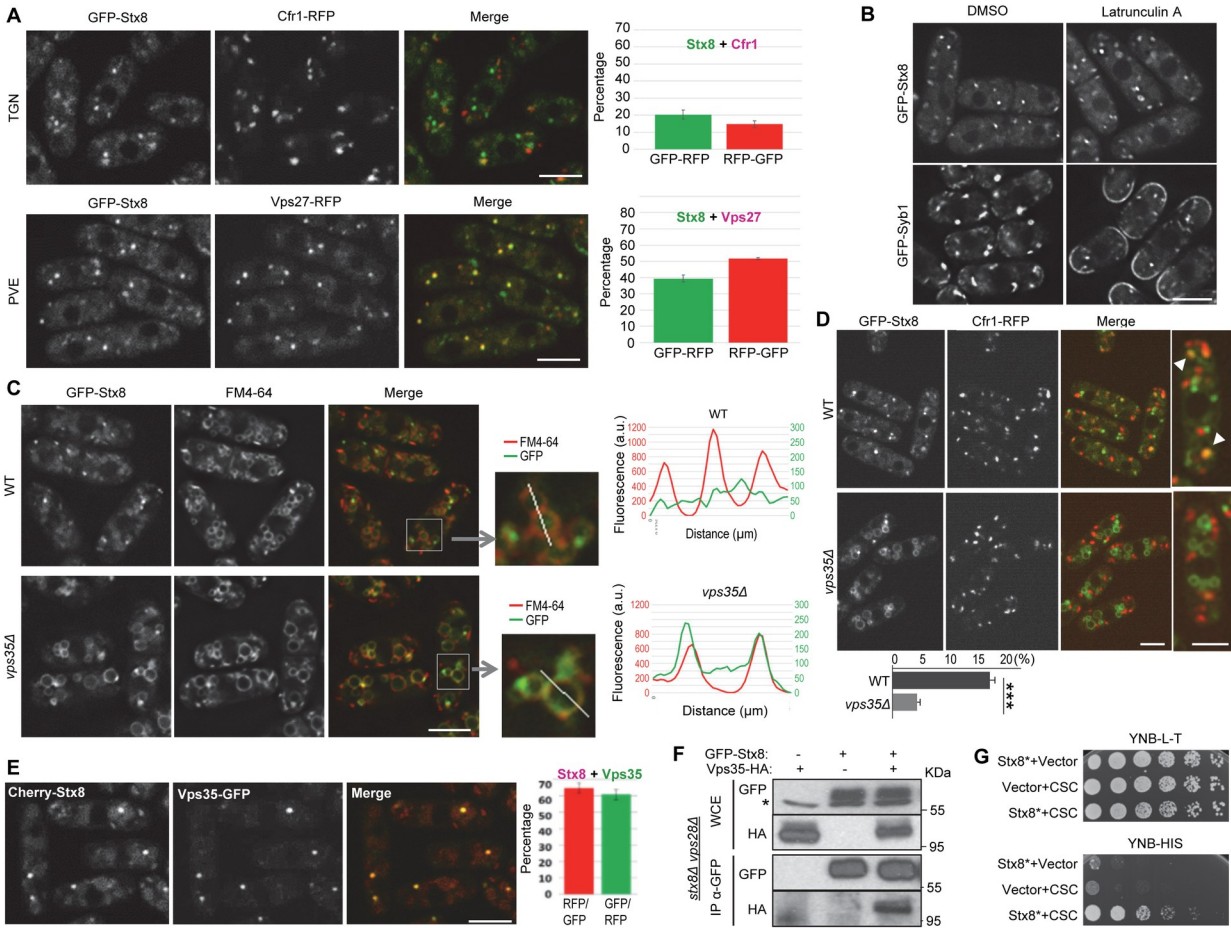

**Fig 3. Stx8 cycles between the *trans*-Golgi network (TGN) and the prevaculolar endosome (PVE) in a retromer-dependent fashion.** (A) Colocalization between GFP-Stx8 and the TGN marker Cfr1-RFP (upper panels) or the PVE marker Vps27-RFP (lower panels). The experiments were performed two times, and a minimum of 400 GFP-Stx8 dots were scored in each experiment. The mean and standard deviation of the results are shown (right panels). For better accuracy, both the percentage of GFP-Stx8 dots that colocalized with Cfr1-RFP or Vps27-RFP dots and the percentage of Cfr1-RFP or Vps27-RFP dots that colocalized with GFP-Stx8 dots were determined. (B) Wild type (WT) cells bearing GFP-Syb1 or GFP-Stx8 were treated with dimethyl sulfoxide (DMSO; solvent) or latrunculin A for 20 minutes and photographed. (C) Distribution of GFP-Stx8 in WT and *vps35Δ* cells. GFP-Stx8 colocalization with FM4-64 is shown. Enlargement of representative vacuoles from the same strains are shown in the central panels. The right panels show line-scans of GFP and FM4-64 fluorescence intensities (a.u., arbitrary units) across the vacuoles, as indicated in the enlargements. (D) Colocalization between GFP-Stx8 and Cfr1-RFP in WT and *vps35Δ* strains. The right panels show enlarged cells. Arrowheads in the WT denote colocalization. The mean, standard deviation and statistical significance (Student's t-test. $P_{value}$ 0.0002) of the colocalization percentages from three independent experiments is shown in the lowest panel. (E) Colocalization between Cherry-Stx8 and Vps35-GFP. The mean and standard deviation of the results are shown (right panels). For better accuracy, both the percentage of Cherry-Stx8 dots that colocalized with Vps35-GFP dots and the percentage of Vps35-GFP dots that colocalized with Cherry-Stx8 dots were determined. Images are single planes captured with by confocal spinning disk microscopy (A, C, D and E) and with a DeltaVision system (B). Scale bar, 5 μm. (F) Stx8 and Vps35 coimmunoprecipitate. Cell extracts from *stx8Δ vps28Δ* strains carrying GFP-Stx8 and/or Vps35-HA were analyzed by western blot using anti-GFP or anti-HA monoclonal antibodies before (WCE, whole-cell extracts) or after immunoprecipitation (IP) with a monoclonal anti-GFP antibody. The asterisk denotes an unspecific band. (G) Serial dilutions of the *Saccharomyces cerevisiae* host strain AH109 strain transformed with empty plasmids (vector) and/or with plasmids expressing Stx8* (Stx8 lacking the TM helix) and/or a fusion Vps29-Vps35-Vps26 protein (CSC) were spotted on yeast nitrogen base medium without leucine and tryptophan (YNB-L-T) and without histidine (YNB-H) and incubated at 30°C for four days. Growth on YNB-H shows direct interaction between Stx8* and CSC.

contributed to the Stx8-Vps35 interaction. Moreover, bimolecular fluorescence complementation (BiFC) showed that Stx8 interacts with Snx3 (Fig 4C), confirming that Stx8 is a Snx3-retromer cargo.

Next, we aimed at identifying the Stx8 region involved in its recycling by retromer. In silico analysis determined that Stx8 is composed of several domains (Fig 4D and Pombase, https://

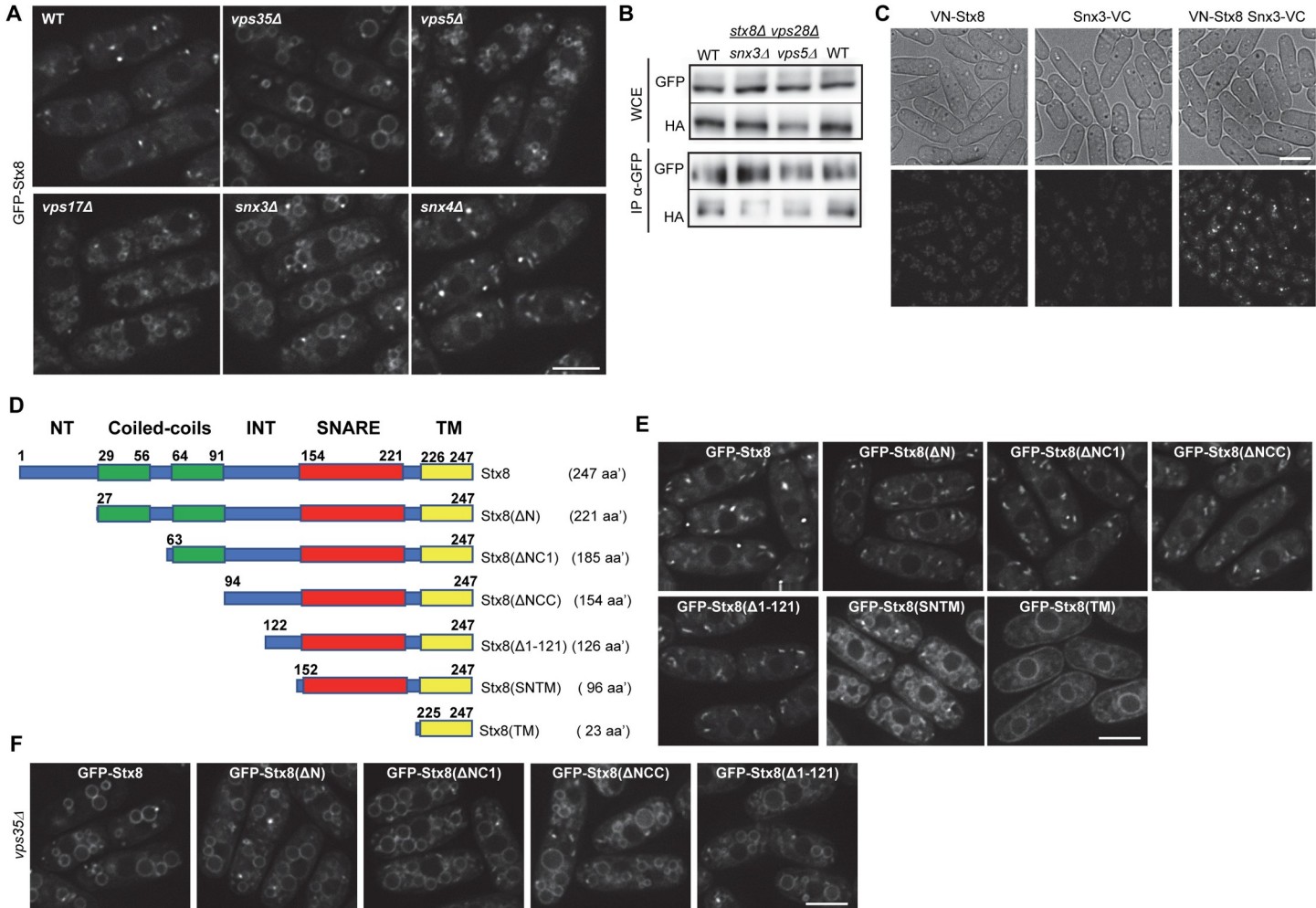

**Fig 4. Requirements for Stx8 recycling from the prevacuolar endosome (PVE) by retromer.** (A) GFP-Stx8 localization in the indicated strains. (B) Coimmunoprecipitation between Stx8 and Vps35 is reduced in the *snx3Δ* and *vps5Δ* mutants. Cell extracts from *stx8Δ vps28Δ* (WT), *stx8Δ vps28Δ snx3Δ* and *stx8Δ vps28Δ vps5Δ* strains expressing GFP-Stx8 and Vps35-HA were analyzed by western blot using anti-GFP or anti-HA monoclonal antibodies before (WCE, whole-cell extracts) or after immunoprecipitation (IP) with a monoclonal anti-GFP antibody. (C) Bright field (upper panels) and fluorescence (lower panels) microscopy of strains that express Stx8 and/or Snx3 fused to the N-terminal (VN) or C-terminal (VC) halves of Venus YFP, respectively. (D) Schematic, not-to-scale representation of different Stx8 protein variants. The unstructured regions are depicted in blue; coiled-coils with no similarity to SNARE domains are depicted in green, the SNARE domain in red, and the transmembrane helix (TM) in yellow. INT (intermediate), denotes an unstructured region delimited by the coiled-coils and the SNARE domain. Bold numbers denote the initial and terminal amino acidic position of each domain. Numbers in parentheses indicate the length of the corresponding protein; aa' indicates amino acids. (E) Localization of the indicated Stx8 variants on a wild type (WT) background. (F) Localization of the indicated Stx8 variants in *vps35Δ*. Images are single planes captured with a DeltaVision system (A, E and F) and a confocal spinning-disk (C). Scale bar, 5 µm.

www.pombase.org). An N-terminal unstructured region (termed NT in the remainder of the text and in the figures) is followed by two consecutive coiled-coil domains separated by 8 amino acids, an internal unstructured region (INT), the SNARE domain, and a C-terminal transmembrane helix (TM). To determine the Stx8 sequence required for its recycling by the Snx3-retromer, we generated several truncations that eliminated N-terminal regions of various sizes (Fig 4D and S1 File). Stx8(ΔN) lacks the 26 most N-terminal amino acids; Stx8(ΔNC1) lacks the first 62 amino acids, which include the N-terminal coiled coil; Stx8(ΔNCC) lacks 93 amino acids, including both N-terminal coiled coils; Stx8(Δ1–121) lacks the N-terminal half of the protein; Stx8(SNTM) bears the SNARE and TM domains; and Stx8(TM) only includes the transmembrane helix. We fused these variants to GFP and expressed them in cells that bore

the native *stx8⁺* allele to visualize the variants that might be nonfunctional and, therefore, would produce a defect in the PVE (Fig 1A and 1F).

Observation under the microscope revealed Stx8(ΔN), Stx8(ΔNC1), Stx8(ΔNCC), and Stx8 (Δ1–121) localized in discrete cytoplasmic dots; Stx8(SNTM) localized in cytoplasmic dots and the vacuole surface; and Stx8(TM) distributed to the endoplasmic reticulum (Fig 4E). To investigate whether the truncations that were observed only in cytoplasmic dots exited the TGN, we analyzed their localization in the *vps35Δ* strain. In the absence of retromer, these truncated variants localized in cytoplasmic dots and the vacuole surface (Fig 4F). These data showed that in the WT they exited the TGN and were recycled back in a retromer-dependent fashion. These results, together with the fact that the Stx8(SNTM) variant was not recycled in the WT, indicated that the C-terminal half of the INT domain bore sequences required for recycling.

## Residues at positions 132–141 are required for Stx8 recycling from the PVE

The results described above strongly suggested that Stx8 retrograde trafficking depended on a protein region between amino acids 122 and 151, included in the INT domain. Nevertheless, it was also possible that the reduced size of Stx8(SNTM) altered its structural conformation and avoided its recycling, similar to how the small Stx8(TM) protein did not exit the endoplasmic reticulum. To distinguish between these possibilities, we created a variant protein that only lacked residues 122–151 (Fig 5A). We observed Stx8(Δ122–151) at the vacuolar membrane (Fig 5B), which confirmed that the recycling of this SNARE from the PVE depended on a 30-residue sequence included in the C-terminal half of the INT domain. The fact that Stx8 coimmunoprecipitation with Vps35 was greatly reduced when it was performed using GFP-Stx8(Δ122–151) instead of GFP-Stx8 (Fig 5C) confirmed the results obtained by microscopy, and demonstrated that amino acids between positions 122 and 151 were required for Stx8 interaction with retromer.

To narrow the interaction zone, we generated three new variants in which series of 10 amino acids were substituted by alanine residues (Fig 5D). We named these proteins Stx8 (122–131)A, Stx8(132–141)A, and Stx8(142–151)A. Analysis of their localization showed that 51% of the cells bearing GFP-Stx8(122–131)A exhibited fluorescence in cytoplasmic dots, and 49% of the cells exhibited fluorescence in cytoplasmic dots and a mild fluorescence in the vacuoles (Fig 5E). In the case of cells bearing GFP-Stx8(132–141)A, there was strong fluorescence in cytoplasmic dots and the vacuole surface in all the cells. Finally, GFP-Stx8(142–151)A fluorescence was similar to that of GFP-Stx8. These results indicated that Stx8 retrograde trafficking depended on the 10 amino acids at positions 132–141 and that some residue at positions 122–131 might play a minor role in the process. To get more information about the role of these residues in Stx8 interaction with retromer, coimmuno-precipitation between these variants and Vps35 was analyzed in three independent experiments. We found that the degree of interaction between Stx8(122–131)A and Vps35, and between Stx8(142–151)A and Vps35 varied between experiments (S5 Fig). On the other hand, physical interaction between Stx8(132–141) and Vps35 was significantly reduced in all experiments (Figs 5F and S5). Two-hybrid analyses were performed to strengthen these results; we found direct interaction between Stx8 and the retromer CSC when the host strain expressed Stx8* (which lacked the TM helix) or Stx8ΔNCC* (lacking the N-terminal Stx8 moiety and the TM helix), although in the latter case interaction seemed less efficient. On the other hand, no interaction was detected between Stx8(132–141)A* and the retromer CSC (Fig 5G). These results confirmed that residues at positions 132–141 were essential for Stx8 interaction with retromer.

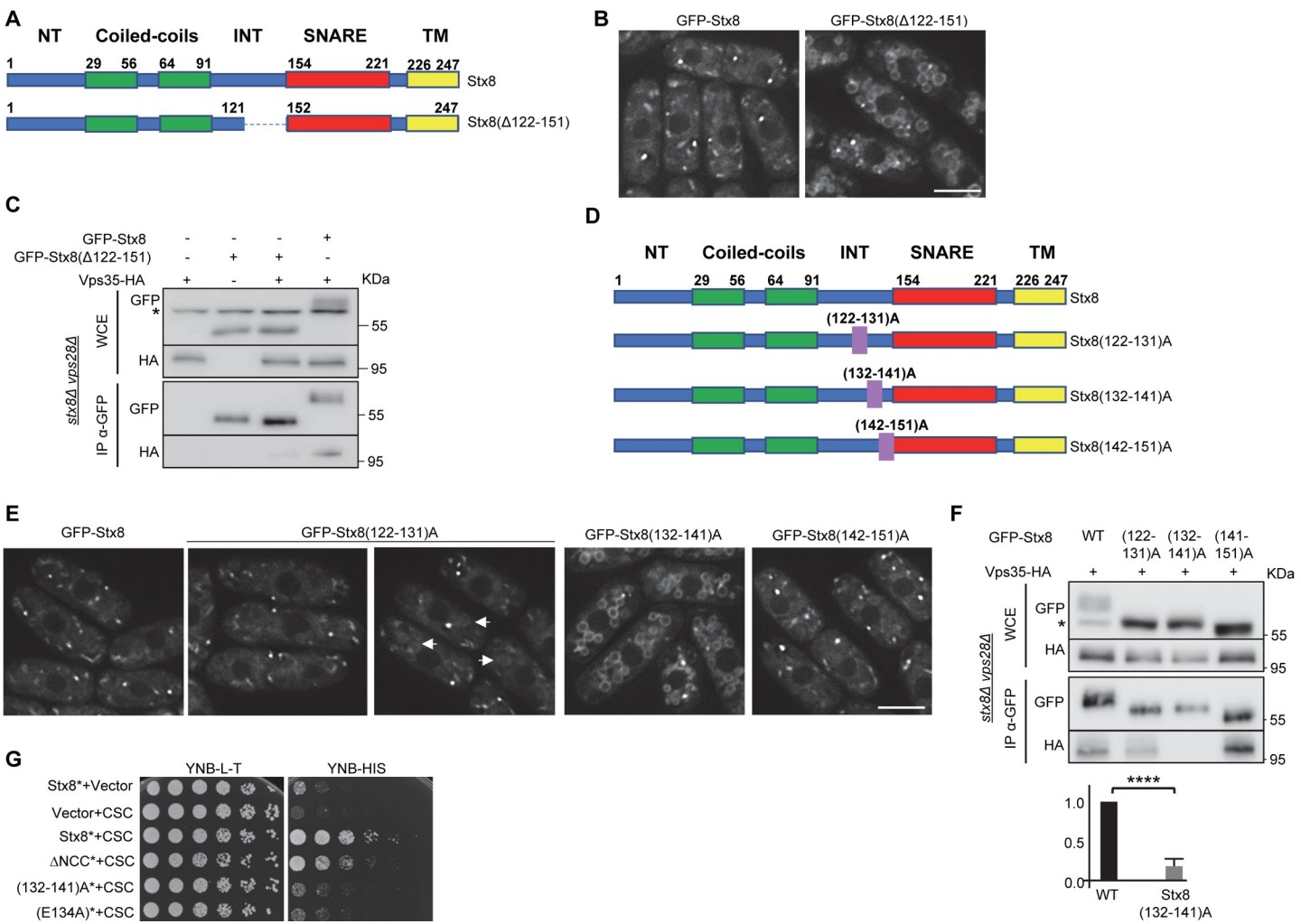

**Fig 5. Residues at positions 132–141 are required for Stx8 recycling by retromer.** (A) Schematic, not-to-scale representation of Stx8 and Stx8(Δ122–151). Bold numbers denote the initial and terminal amino acid position of the regions of interest. The dashed line represents the protein region deleted. (B) Localization of the indicated proteins. (C) Coimmunoprecipitation between Stx8(Δ122–152) and Vps35. As a control, the interaction between Stx8 and Vps35 was included (lanes on the right). Cell extracts from *stx8Δ vps28Δ* strains carrying GFP-Stx8, GFP-Stx8(Δ122–152) and/or Vps35-HA were analyzed by western blot using anti-GFP or anti-HA monoclonal antibodies before (WCE, whole-cell extracts) or after immunoprecipitation (IP) with a monoclonal anti-GFP antibody. The asterisk denotes a nonspecific band. (D) Schematic, not-to-scale representation of the indicated Stx8 variants. The numbers in parentheses indicate the residues that were mutated to alanine, and the purple rectangles indicate their relative position in the protein. (E) Localization of the indicated Stx8 variants. Arrows denote vacuoles with mild fluorescence at their surface. (F) The same details as in C, but coimmunoprecipitation between the indicated GFP-Stx8 variants and Vps35-HA was analyzed. The amount of Vps35-HA that coimmunoprecipitated with GFP-Stx8 (WT) and GFP-Stx8(132–142)A proteins was estimated and quantified (see Materials and methods). The statistical significance of the differences is shown in the lowest panel (t- test; ****, p<0,0001). G. Serial dilutions of the *Saccharomyces cerevisiae* AH109 host strain transformed with empty plasmids (vector) and/or with plasmids expressing Stx8* (Stx8 lacking the TM helix), the indicated Stx8* variants and/or a fusion Vps29-Vps35-Vps26 protein (CSC) were spotted on yeast nitrogen base medium without leucine and tryptophan (YNB-L-T) and without histidine (YNB-H) and incubated at 30˚C for four days. Growth on YNB-H shows direct interaction between Stx8* and CSC. In B and E, images are single planes captured with a DeltaVision system. Scale bar, 5 μm.

## Residues $^{133}$I, $^{134}$E, $^{135}$M, and $^{138}$M are essential for Stx8 recycling

To pinpoint the amino acids responsible for Stx8 recycling, we performed comprehensive alanine-scanning mutagenesis [57] on the DIEMEAMIPV residues, at positions 132–141 (Fig 6A). We analyzed the localization of the resulting Stx8 variants by microscopy. As shown in Fig 6B, there was neat GFP fluorescence at the vacuolar surface of strains bearing Stx8(I133A), Stx8(E134A), Stx8(M135A), and Stx8(M138A). These results showed that $^{133}$I, $^{134}$E, $^{135}$M, and

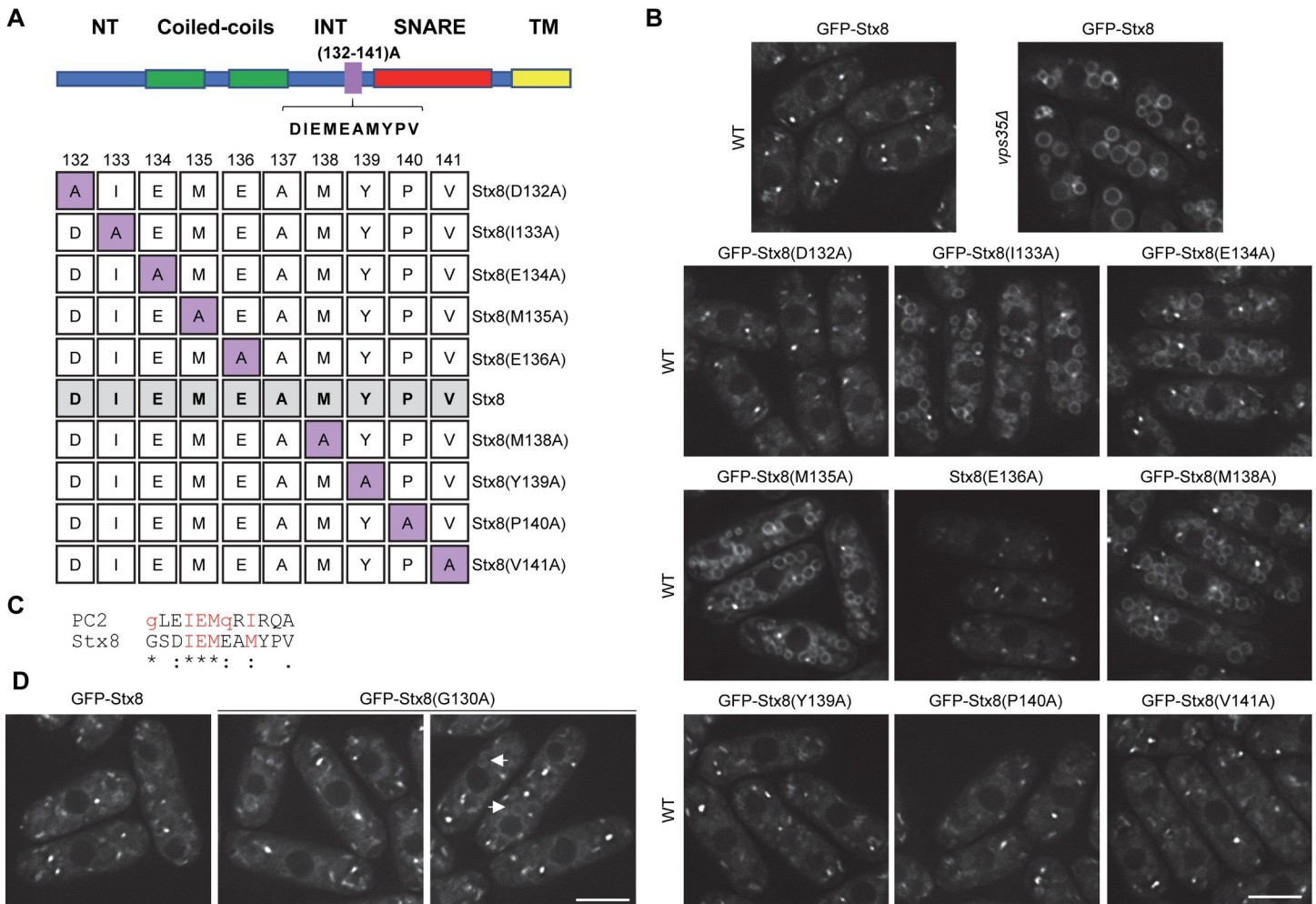

**Fig 6. Residues [133]I, [134]E, [135]M and [138]M are essential for Stx8 recycling from the prevaculoar endosome (PVE).** (A) Upper panel: Sequence and relative position of the residues subjected to alanine-scanning mutagenesis. Lower panel: Sequence and name of the Stx8 variants produced by the mutagenesis. The numbers at the top indicate the position of the residue that was mutated. The lane denoted as Stx8, shaded in grey, corresponds to the native protein, where the residue at position 137 is an alanine. Purple squares indicate the residues mutated to alanine. (B) Distribution of the indicated Stx8 variants. The localization of native Stx8 in wild type (WT) and *vps35Δ* backgrounds are shown for comparison (upper panels). (C) Alignment of Stx8 and human polycystin-2 (PC2) motifs involved in protein recycling by retromer. The residues that participate in the process are in red font, and essential residues are in capital letters. (D) Localization of Stx8(G130)A in WT cells. Arrows denote vacuoles with mild fluorescence at their surface. Images are single planes captured with a DeltaVision system. Scale bar, 5 μm.

[138]M were all essential for the recycling of this SNARE from the PVE. In agreement with this, Stx8E(134)A failed to interact with the retromer CSC (Fig 5G).

The IEMeaM motif resembles the polycystin-2 (PC2) GxxIEMQxIx motif that had been deduced after comparing several sequences, and mediates the interaction between human PC2 and Vps35 and Vps26 [40,58]. The Stx8 sequence includes a glutamine at position 130 (Fig 6C). Since the glycine at the equivalent position in the human PC2 has been suggested to participate in its recycling by retromer, we mutated this glycine to alanine to generate Stx8 (G130A). Microscopy showed that this variant distributed in cytoplasmic dots in 67% of the cells, while 33% of the cells exhibited fluorescence in dots and a mild fluorescence in at least one vacuole (Fig 6D). This finding was in agreement with the fact that some cells bearing GFP-Stx8(122–131)A exhibited mild fluorescence at the vacuole surface (Fig 5E).

To understand the contribution of Stx8 recycling by retromer to the cell physiology, we investigated the capacity of some of these Stx8 variants to complement the *stx8Δ* sensitivity to K⁺. To do so we crossed the *stx8Δ* null mutant with the strains bearing GFP-Stx8, GFP-Stx8 (ΔSN), GFP-Stx8(132–141)A, and some of the point mutants (I133A, E134A and M135A). All the *stx8Δ* GFP-Stx8 clones analyzed were resistant, as the WT control, and all the *stx8Δ* GFP-Stx8(ΔSN) clones were sensitive, as the *stx8Δ* control (S5 Fig). Surprisingly, in the case of *stx8Δ* GFP-Stx8(132–141)A, *stx8Δ* GFP-Stx8(I133A), *stx8Δ* GFP-Stx8(E134A), and *stx8Δ* GFP-Stx8(M135)A we found differences in the sensitivity between clones (See some examples in the S5 Fig). This variability suggests that there is either a suppressor or an enhancer of the salt sensitivity phenotype when Stx8 recycling is compromised. Nevertheless, the sensitivity is never as drastic as that shown by cells bearing a deletion in the SNARE domain (Fig 2B and S6). These results suggest that once the SNARE has mediated vesicle fusion to the target membrane at the PVE, its retromer-mediated recycling is not essential. This recycling might contribute to full efficiency, but the cell would be able to cope with a reduced recycling. The strong potassium sensitivity exhibited by the *vps35Δ* mutant (Fig 2A) must be produced by the defective recycling of Stx8 and other proteins.

## Role of the SNARE domain in Stx8 recycling

The results described above showed that residues at positions 133, 134, 135, and 138 are essential for Stx8 recycling. Nevertheless, these results did not rule out the possibility that other residues in the C-terminal half of the protein participate in this recycling. Specifically, residues in the SNARE domain might participate in the process. To address this issue, we constructed a variant protein that lacked this domain (Fig 7A). Observation of GFP-Stx8(ΔSN) under the microscope showed that it localized in cytoplasmic dots and the vacuole surface (Fig 7B). Although the vacuolar signal was not as neat as that of GFP-Stx8 in *vps35Δ* cells, this result showed that the SNARE domain participated in Stx8 recycling. To determine the residues in this domain implicated in the process, we generated two new variants that lacked either the N-terminal or the C-terminal halves of the SNARE domain (Fig 7C). These Stx8 variants were termed Stx8(Δ152–187) and Stx8(Δ188–224), respectively. To our surprise, none of these variants was retained at the vacuole surface (Fig 7D). To determine whether they exited the TGN, we analyzed their distribution in a *vps35Δ* mutant. Both proteins were observed at the vacuolar membrane; hence, in the WT background they exited the TGN and were recycled by retromer.

The fact that, according to informatics, both halves of the SNARE domain can form coiled-coils (S2 File) suggests that this structure might be important for the functionality of the neighboring sorting motif. Nevertheless, it was also possible that the IEMeaM motif and the TM domain have to be separated by a minimum number of amino acids, with or without a structure. To discern between these possibilities, we replaced the SNARE domain with a string of 36 alanine and glycine residues (Fig 7E); these residues would serve as a hinge without conferring an α-helical structure. Observation under the microscope showed that GFP-Stx8(A+G) localized in cytoplasmic dots. There was also fluorescence in the endoplasmic reticulum, but not at the vacuole surface (Fig 7F). On the contrary, there was GFP-Stx8(A+G) fluorescence at the vacuole surface of *vps35Δ* cells. This result demonstrated that the presence of a linker sequence between the sorting motif and the TM helix—but not the sequence itself—was required for Stx8 retrograde trafficking.

## Presence of an IEMeaM motif in other SNAREs

As mentioned above, the Stx8 sorting motif resembles that of PC2. The resulting consensus sequence is IEMxx(I/M), where each of the IEM and I/M residues is essential for recycling by/

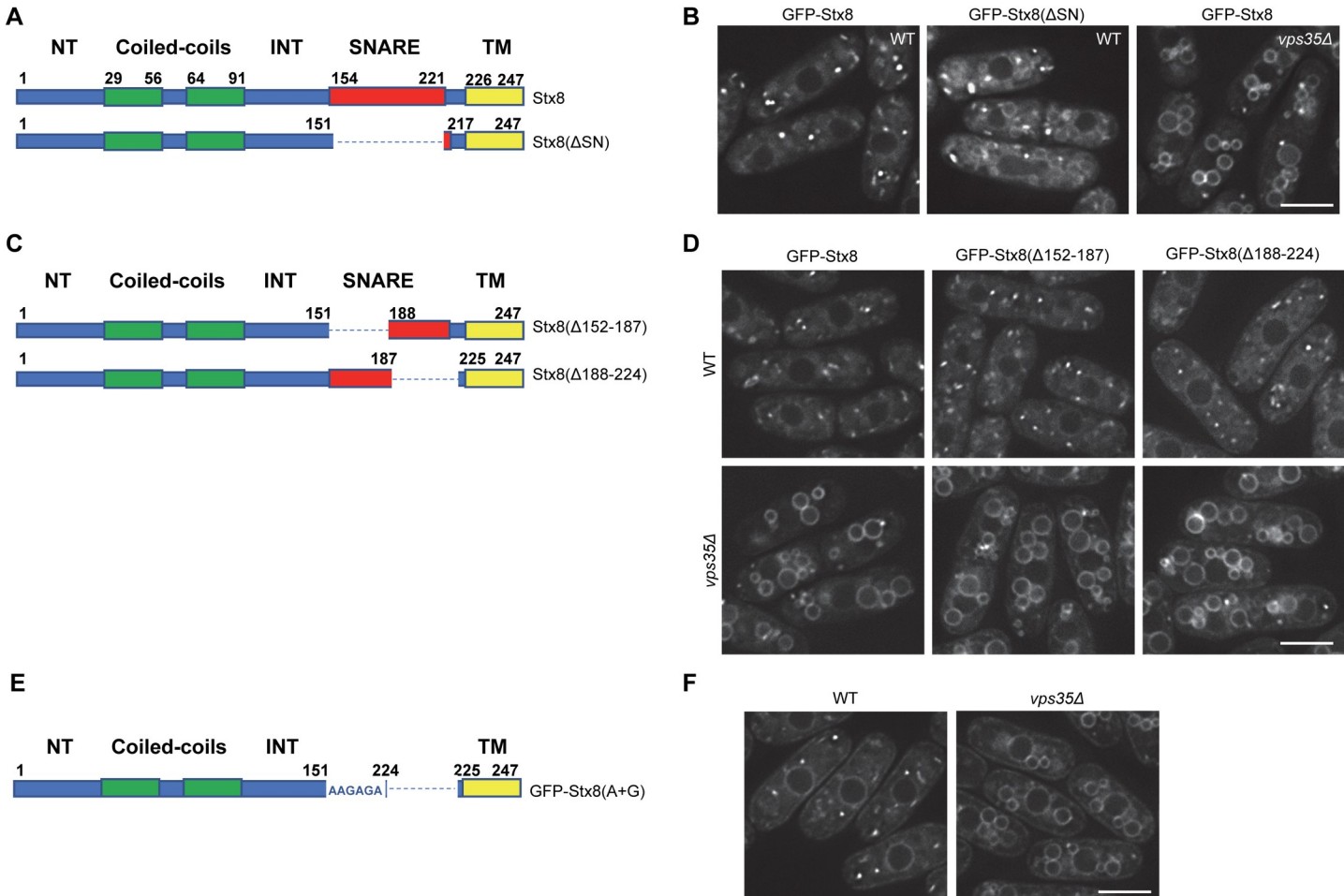

**Fig 7. Contribution of the SNARE domain to Stx8 recycling.** (A) Schematic, not-to-scale representation of Stx8 and Stx8(ΔSN), lacking the SNARE domain. (B) Distribution of Stx8 and Stx8(ΔSN). (C) Schematic, not-to-scale representation of Stx8(Δ152–187) and Stx8(Δ188–224) variants, lacking the residues indicated in parentheses. (D) Distribution of the indicated Stx8 variants in wild type (WT) and *vps35Δ* cells. (E) Schematic, not-to-scale representation of Stx8(A+G), where residues 152–223 were changed to a string of alanine and glycine residues. (F) Distribution of Stx8(A+G) in WT and *vps35Δ* cells. In all the protein representations, bold numbers denote the initial and terminal amino acidic position of the regions of interest, and the dashed lines represent the protein region missing in the corresponding variant. Images are single planes captured with a DeltaVision system. Scale bar, 5 μm.

interaction with retromer, and x is any residue. This motif is different from those described for other retromer cargoes [26,29,36–39]. Nevertheless, it marginally fits the [+-]ΨΦΨ(L/M) consensus described by Lucas et al. [26]. According to their nomenclature, [+-] is a charged residue, Ψ is a hydrophobic residue having a large aliphatic side tail, and Φ is an aromatic residue. Given that the normalized nomenclature is to denote all hydrophobic residues with Φ and aromatic residues with Ω [59], we will refer to this motif as [+-]ΨΩΨ(L/M). In the case of the IEMxx(I/M) motif, the [+-] position would correspond to E (a negatively charged residue) (Fig 8A), Ψ would correspond to M, ΩΨ would correspond to xx, and (L/M) would correspond to (I/M). I, L, and M are hydrophobic residues with large aliphatic side chains, and thus they can be denoted by Ψ. Consequently, the Stx8/PC2 sorting motif would fit into an IEMxxΨ consensus.

To investigate whether the IEMxxΨ motif is conserved, we used BLASTP [60] and ClustalW [61] to identify proteins annotated as Fsv1 in other *Schizosaccharomyces* species (S3 File), and to align them with Stx8. The results showed the presence of this motif in the *S. pombe, S.*

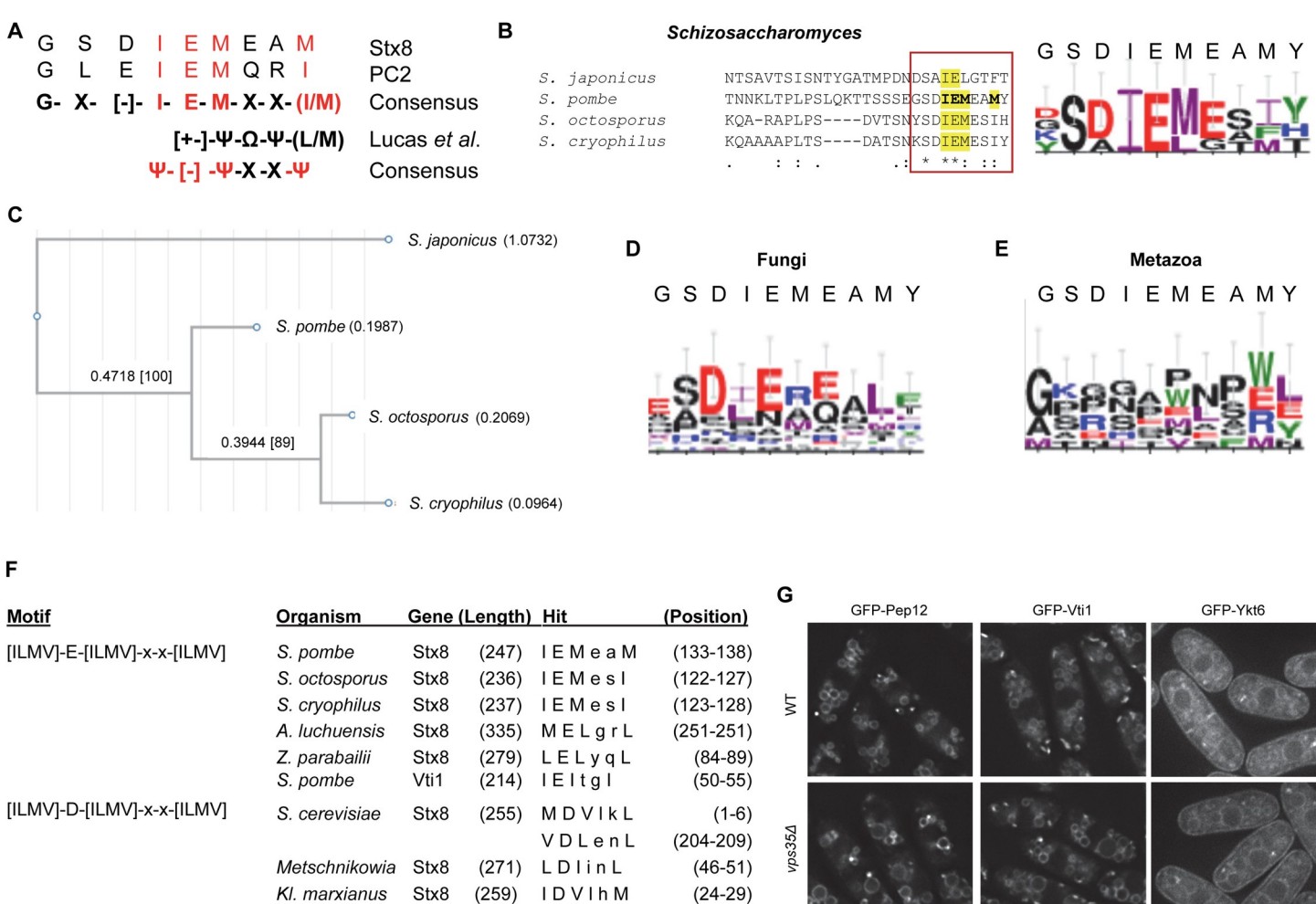

**Fig 8. Conservation of the Stx8 IEMeaM motif.** (A) Comparison between the motifs in *Schizosaccharomyces pombe* Stx8 and human PC2, and the consensus motif defined by Lucas et al [26] where Ψ was hydrophobic residue with large aliphatic chain; [+-] a charged residue; and Ω a bulky hydrophobic (aromatic) residue [26]. In the case of Stx8 and PC2, where the charged residue is an E, the consensus at this position is [–]. The residues essential for function are in red font. (B) Left panel: BLASTP alignment of Fsv1/Stx8 proteins from the *Schizosaccharomyces* species. The square denotes the region where the IEMeaM motif is located. Right panel: Weblogo showing the conservation of the residues denoted by the square in the left panel. The *S. pombe* sequence is indicated at the top. E, D in red; H, K, and R in blue; F, Y, and W in green; and I, L, M, and V in purple. (C) Phylogram of the Fsv1/Stx8 sequences aligned with clustal W (B). (D) The same details as in (B), but the analysis was extended to 19 fungal Stx8 sequences. (E) The same details as in (B), but the Weblogo was deduced from the *S. pombe*, *Homo sapiens*, *Pan troglodytes*, *Rattus norvegicus*, *Mus musculus*, *Danio rerio*, *Drosophila Melanogaster*, and *Arabidopsis thaliana* Syntaxin 8 sequences. (F) ScanProsite hits detected by scanning for the indicated motifs. (G) Localization of the indicated SNAREs in WT and *vps35Δ*.

*octosporus*, and *S. cryophilus* proteins, and a related motif in Fsv1 from the most distant species, *S. japonicus*. In the latter, there is an L in the M position, and the last residue is an aromatic residue (Fig 8B). The phylogenetic tree of these motifs showed that their divergence matched that of the *Schizosaccharomyces* species (Fig 8C, and [62,63]). When we extended the comparison to proteins from other fungal species, the degree of conservation was lower (Fig 8D). Finally, when we compared Stx8 with syntaxin 8 from metazoans (*Homo sapiens*, *Pan troglodytes*, *Rattus norvegicus*, *Mus musculus*, *Danio rerio*, *Drosophila melanogaster*) and from *Arabidopsis thaliana*, there was no consensus regarding the identity or the nature of the residues in equivalent positions (Fig 8E). These results showed that the IEMxxΨ motif is conserved in Stx8/Fsv1 from species that are closely related to *S. pombe*.

Using a different approach, we searched a loose Ψ[−]ΨxxΨ motif in several proteins annotated as syntaxin 8 (S4 File). We screened these proteins with ScanProsite [64] to detect [ILMV]-[ED]-[ILMV]-x-x-[ILMV]. The scan produced five hits with an E at the [ED] position (*S. pombe*, *S. octosporus*, *S. cryophilus*, *Aspergillus luchuensis*, and *Zygosaccharomyces parabailii*; Fig 8F), and four hits with a D (*Saccharomyces cerevisiae*, *Metschnikowia* sp., *Kluyveromyces marxianus*, and *D. melanogaster*; Fig 8F). We did not detect human syntaxin 8. We also screened *S. pombe* Vti1, Ykt6, and Pep12 (the Stx8 SNARE partners at the PVE), whose regulation by retromer is difficult to determine because they are normally observed at the vacuole membrane (Fig 8G). There was a hit only for Vti1 (⁵⁰IEItgI⁵⁵). When we screened *S. cerevisiae* Pep12, which depends on the Snx3-retromer for recycling from the PVE [31], there was no hit. Still, there were no hits when we screened Pep12 for [ILMV]-[ED]-[ILMV]-x-x-[FYW] (Ψ+ΨXXΩ), which detected *S. japonicus* Fsv1/Stx8. Moreover, Pep12 does not contain ΨxΩxΨ, ΩXΩXΩ, or ΩΩ motifs, which bear aromatic residues at various positions. When we screened the mammalian syntaxin 8 sequences for all these motifs, there were only ΩΩ hits in several homologues, including human (⁶WF⁷). The relevance of these motifs in their regulation is unknown.

In summary, these results showed that a ΨEΨxxΨ motif is present in some but not all the Stx8 homologues. Additionally, the analysis of Pep12 showed that additional sorting motifs must be recognized by this protein complex.

## Presence of an IEMxxΨ motif in other polycystin-2 proteins

The fact that the Stx8 retromer sorting motif had undergone divergence along the evolution of the genus *Schizosaccharomyces*, but was not conserved in distantly-related fungi or mammalian syntaxin 8 homologues prompted us to investigate the conservation/divergence of this motif in polycystin-2 sequences. To do so, we used Scanprosite to detect possible [ILMV]-[EDHKR]-[ILMV]-X-X-[ILMV] loose motifs in several metazoan, including different *Drosophila* and *Caenorhabditis* species (S5 File). We found that the GLEIEMERI motif was conserved in mammals, with the E at position 7 being the only variable residue. At this position, a Q instead of an E is present in 50% of the 16 sequences analyzed (S5 File and Fig 9A). The motif was more variable in fish; we compared nine sequences, including that of *Danio rerio*, which was analyzed by Tilley et al., (2018). The comparison of the motifs produced by the scan led to an XEG(I/L)EMEX(I/M) consensus motif, with an I at position 4 in eight sequences and an I at position 9 in seven sequences (S5 File and Fig 9B).

The analysis of PC2 sequences from four *Caenorhabditis* species produced hits that led to a loose XXΦΨ[−]ΨxxΦ motif (Fig 9C). The motifs in the most closely-related species (*C. brigsae* and *C. remanei*) are identical (WNLVDVALL), while those in *C. brenneri* CB5161(NENIEMIEL) and *C. elegans* (RSFMEVGGY) are more divergent. The scan produced hits in the five *Drosophila* sequences that we analyzed (S5 File), and their comparison led to a loose XL[+]Ψ[+]L[+]X(L/M) motif (Fig 9D). The motifs in *C. elegans* (RSFMEVGGY) and *D. melanogaster* (NLRLRLRTL) PC2 proteins are very different from that in *H. sapiens* (GLEIEMQRI), in agreement with the divergence of the PC2 N-terminal regions reported by Tilley et al. (2018) for these species.

We also analyzed the *Schizosaccharomyces* species (Fig 9E); the comparison of the hits produced by Scanprosite led to a loose ΦXXI[+-]ΨXXΨ motif, with the sequences from the closely-related species *S. octosporus* and *S. cryophilus* being identical (MGIIKIGFM) and those from *S. japonicus* (FLAIKLPIV) and *S. pombe* (AYGIEIKKV) more distant. Finally, when we compared the sequences for all organisms at one time, the most represented residues led to a XXXΨ[+-]ΨXXΨ consensus (Fig 9F).

The results shown in Figs 8A–8G and 9A–9E are preliminary because, i) we have analyzed the potential sorting motifs in only two proteins (syntaxin 8 and polycistin-2 homologues); ii)

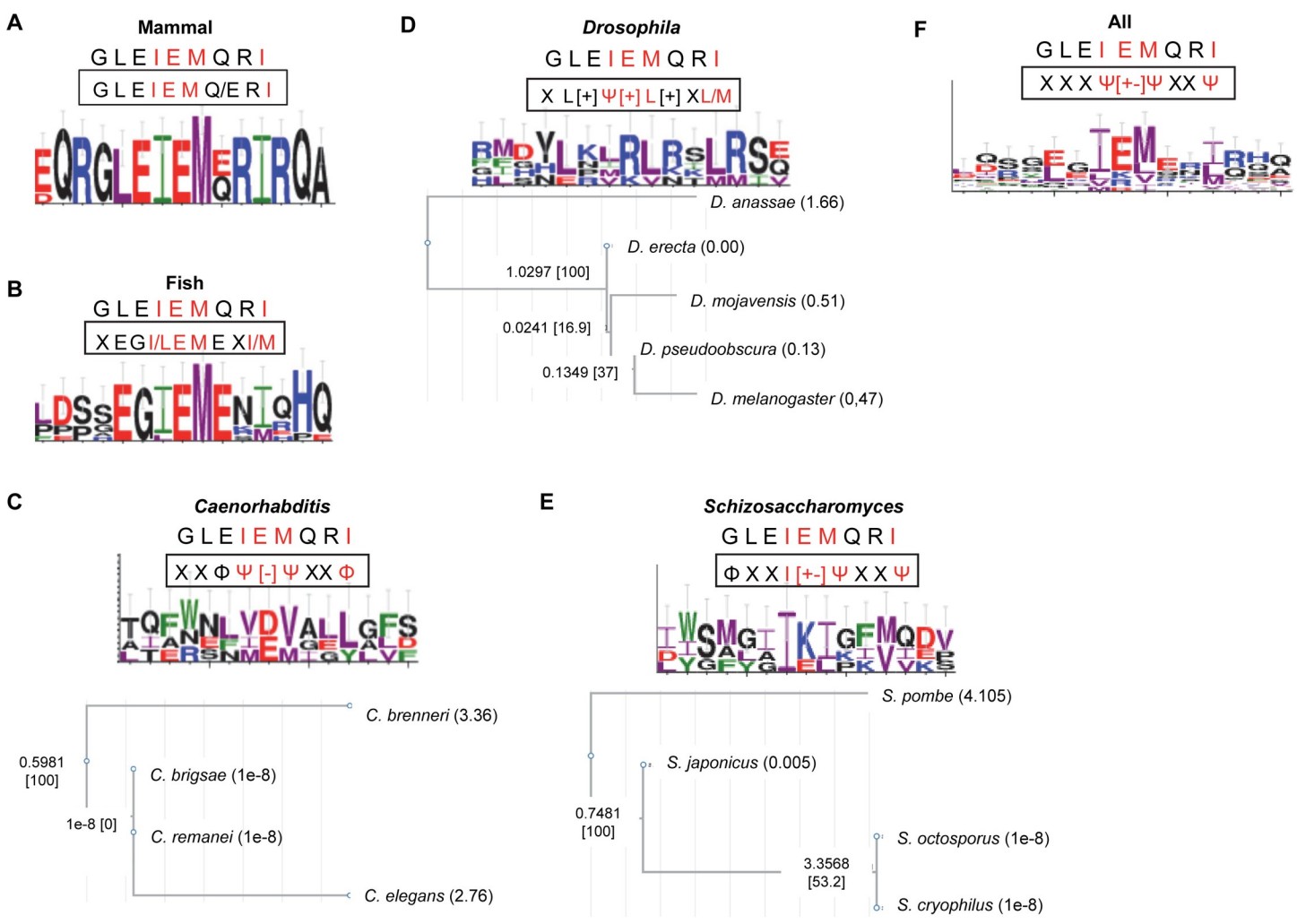

**Fig 9. Conservation of the retromer sorting motif in PC2 sequences from different organisms.** The panels show Weblogos that indicate the conservation of the residues in equivalent positions of the N-terminal regions of polycystin-2 (PC2) sequences retrieved from Uniprot and NCBI databases (see the S5 File for details). The PC2 sequences from Mammal (A), Fish (B), *Caenorhabditis* (C), *Drosophila* (D), *Schizosaccharomyces* (E), and from all the organisms under study (F) were compared using ClustalW before the analysis. In each panel, the human PC2 sequence that includes the sorting motif is shown at the top, and the group consensus is shown into a rectangle. E, D in red; H, K, and R in blue; F, Y, and W in green; and I, L, M, and V in purple. The phylogeny of *Drosophila*, *Caenorhabditis* and *Schizosaccharomyces* sequences is shown in the lower C, D, and E panels, respectively.

the sequences are from a limited number of organisms; iii) in some organisms there are several isoforms of these proteins; iv) in most cases, these proteins have not been shown to be regulated by retromer. Nevertheless, they indicate certain degree of evolutionary divergence of the retromer sorting motif into each group of organisms, and between groups. On the other hand, they support the notion that the presence of almost-identical motifs in *S. pombe* Stx8 and mammalian PC2 might be a case of convergent evolution.

## Discussion

### Stx8, functionality of the PVE, and ion homeostasis

According to sequence homology, the *S. pombe* pep12[+](Qa), vti1[+] (Qb), stx8[+] (Qc), and ykt6[+] (R) would constitute a SNARE involved in homotypic fusion at the PVE [65–70]. vti1Δ and ykt6Δ mutants are inviable and the viability of pep12Δ is controversial [35,65,67,71]. Analysis

of a *pep12Δ* mutant indicated that *pep12*[+] is related to vacuolar homotypic fusion [71]. *fsv1/stx8Δ* cells are viable, and Fsv1/Stx8 plays a role in vesicle trafficking at the PVE ([34,35]; this work). Some SNARE interactions are promiscuous, at least in vitro [72]; therefore, the reason why *stx8Δ* cells grew efficiently, while cells with deletions for its partner SNAREs did not, could be that another SNARE took up its function. Although *vsl1*[+] is a Qc SNARE that exerts its function at the PVE-vacuole interface, it did not undertake the Fsv1/Stx8 function at the PVE (S3 Fig; [34]). An alternative explanation would be that while the function of Pep12, Vti1, and Ykt6 would be relevant for several trafficking steps [66,73–75], Stx8 function might be restricted to the PVE-vacuole interface. Stx8, similar to its human counterpart, plays a role in the fusion of PVE-derived vesicles to vacuoles and vacuolar homotypic fusion [7,8,34]. Nevertheless, the fact that vacuoles appeared normal in *fsv1/stx8Δ* cells (Fig 1; [34,35]) suggested that this function might be minor and only evident when *vsl1*[+] is deleted [34]. Stx8 function would be more relevant for PVE homotypic fusion and *S. pombe* Pep12-Vti1-Stx8-Ykt6 SNARE would be equivalent to human Stx7-Vti1b-Stx8-Vamp8, while Pep12-Vti1-Vsl1-Ykt6 would be equivalent to Stx7-Vti1b-Stx8-Vamp7 [7,8]. Although we cannot rule out the hypothesis that another SNARE is partially redundant with Stx8, homotypic fusion at the PVE might not be essential, provided that PVE-to-vacuole fusion is allowed. This eventuality would explain why trafficking defects in *stx8Δ* were mild. Nevertheless, the *stx8Δ* PVE compartments would be small, explaining the reduced fluorescence of the markers. The small size would also imply a greater membrane curvature that might hinder the formation of intraluminal vesicles, leading to a defect in PVE functionality.

Syntaxin 8 has been implicated in the regulation of plasma membrane ion channels [40,76–78] and, we show that Stx8 was required for growth under ionic stress. Thus, regulating ion homeostasis might be a conserved syntaxin 8 function. Our results showed that both the altered PVE and the K[+] sensitivity of *stx8Δ* cells were due to the lack of the SNARE domain; these data showed that both defects were related. These results strongly suggest that ion regulation by mammalian syntaxin 8 might also be exerted at the late endosome.

## An IEMeaM motif is essential for Stx8 recycling by the Snx3-retromer

Our results showed that i) Stx8 cycled between the TGN and the PVE in a retromer-dependent manner; and ii) retromer and Stx8 formed a complex. Retromer is also involved in the recycling of the *S. cerevisiae* SNAREs Pep12 and Snc1 [30,31]. Additionally, recycling of some proteins depends on SNXs other than Vps5/SNX1/SNX2 and Vps17/SNX5/SNX6; while Syb1/Snc1 depends on Snx4, [31,32,79,80], *S. cerevisiae* Pep12 depends on Snx3 [31], which is considered to be a dedicated retromer adaptor [20,27]. We have shown that Stx8 is a cargo of the Snx3-retromer. The reason why a protein is recognized by a specific SNX is not fully understood. Syb1/Snc1 cycles through the plasma membrane while Stx8 does not; nevertheless, the trafficking route would not be the only determinant because Ftr1 depends on Snx3 [27] while Can1 depends on Snx4 [81], and both Can1 and Ftr1 are plasma membrane proteins. Regarding the existence of cargo sorting motifs that are specifically recognized by retromer and/or different SNXs, multiple analyses have been conducted by analyzing the sorting requirements of yeast and mammalian proteins, and by performing in silico predictions based on these requirements [20,26,27,29,36–40,42,58,82–90]. These analyses did not lead to a unique signal that would be recognized by retromer, Snx3, or Snx4. Furthermore, for some retromer and SNX cargoes, no sorting motif has been identified. Nonetheless, these studies have helped to establish some consensus sequences that mediate cargo recognition by retromer and Snx3-retromer. These consensus sequences include several combinations of hydrophobic residues with large aliphatic or aromatic side chains that are contiguous, alternate, or separated by

several residues, leading to a bipartite signal; some examples are $\Omega$X(L/M) [90], $\Phi$X$\Omega$X$\Phi$(X)n$\Phi$ [38], and [+-]$\Psi\Omega\Psi$(L/M) [26]. Stx8 truncation analyses showed that its C-terminal moiety was sufficient for recycling. Motifs fitting some of the consensus sequences mentioned above are present in this part of the protein ([137]AMYPVDGNDPDPINVNVL[154] and [182]MGYAMN-TEL[190]). Nevertheless, we found that these motifs were not required for recycling, while eliminating [133]I, [134]E, [135]M, and [138]M residues halted the process. The IEM block slightly resembles the $\Omega$E$\Omega$ motif of Kex2, Ear1, Ymd8 and Any1 [20], although the hydrophobic residues in these proteins are larger because they bear aromatic chains. The IEM block also resembles the INL ($\Psi$N$\Psi$) motif in Ear1, where I and L are essential for recycling. Nevertheless, this motif is part of the bipartite [453]PPGFEF[458]–[473]INL[475] signal, while the IEMeaM motif is the Stx8 only sorting signal.

If the cargo motif is not very strict, how can retromer recognize it? There are several interactions between the retromer subunits, and some of the residues involved in them contribute to cargo recognition [26,38,39]. These residues are conserved in the human, *S. cerevisiae*, and *S. pombe* homologues (S6 File). Different cargoes interact with different sites at the retromer [39], an eventuality that helps to explain the diversity in sorting motifs, which seem to be variations of a very loose and general binding motif. The determinant for the binding of these residues to the cargo might be a general aspect of their nature, which would permit certain variability. Most of the sorting signals described include hydrophobic residues with either aromatic or large aliphatic chains. It is possible that their large volume facilitates contact with the residues that are involved in recognition. For example, [555]Y in the DMT1-II sorting motif accommodates within a large pocket in the Vps26-SNX3 interface, which contributes to cargo interaction with the Snx3-retromer [26]. It is important to bear in mind that only two or three residues are involved in each contact such that the binding is unstable. Therefore, the presence of several large residues would allow multiple contacts, a phenomenon that explains why mutating single residues in the sorting motif of some retromer cargoes only reduced binding and/or recycling while mutating several residues led to a severe defect [39,83,84,88,90]. Moreover, substituting a residue for a large amino acid is less detrimental than substituting it for a small one, such as alanine [84,86,90]. For Stx8 and PC2, mutating each residue in the motif halts recycling ([40]; this work), in line with the hypothesis that the absence of bulky aromatic residues would lead to inefficient binding with the retromer subunits, which would be destabilized by eliminating each single contact.

Another mutually non-exclusive possibility would be that the binding determinant is a structural feature, which might be achieved by different residue combinations in the cargo motif. Sorting motifs are generally located in unstructured protein regions, which are flexible and would allow the motif to fit into the retromer pocket that contains the binding residues [26,38,39,91]. In the case of CI-MPR, the sorting motif folds into a $\beta$ hairpin that binds SNX5/SNX6 [38]. Reaching a favorable structure of the sorting motif might be influenced by some proximal residues [88,90]. In the case of Stx8, [130]G might contribute to the binding because its mutation led to a partial defect in sorting. The IEMeaM motif is in the INT domain, an unstructured region located between several coiled-coils. Nevertheless, these highly structured regions are not relevant for Stx8 recycling because eliminating the N-terminal coils and replacing the SNARE domain by a string of alanine and glycine residues did not halt Stx8 recycling.

Additionally, the distance between the sorting motif and the TM helix has been shown to be relevant for cargo-adaptor binding. Thus, for clathrin adaptors the sorting motif is at least seven residues from the helix [92]; in the case of DMT1-II, the [555]Y must be more than 15 amino acids away from the TM helix [90], and alterations in the spacing between the $\Omega\Omega$ sequence and the TM domain block CD-MPR recycling [88]. In the case of Stx8, the absence of the SNARE domain blocked recycling while the presence of either the N-terminal half or

the C-terminal half of this domain, or a string of alanine and glycine residues allowed it. These results showed that the presence of several amino acids—not their nature—is important for recycling. The distance between both protein regions might facilitate folding such that the sorting motif is exposed to its binding partners, and/or might provide space to accommodate the retromer/SNXs complex. In this regard, it has been shown that the core retromer subunits Vps26, Vps35 and Vps29 do not contact directly the lipid membrane but are layered on the top of Vps5 oligomers [93]. Therefore, the linker between the retromer sorting motif and the TM helix might be required to guarantee that the residues composing the sorting motif reach the distal retromer subunits. Finally, in the case of Snc1, the presence of lysine residues in and proximal to the TM domain influences the orientation of the helix, allowing the exposure of a WY signal [79,94]. The spacing between the sorting motif and the TM helix might favor an adequate helix orientation, which would contribute to cargo binding.

### Conservation of the Snx3-retromer sorting motif

The IEMeaM sorting motif in Stx8 is similar to the GxxIEMQxI consensus defined for polycystin-2 proteins [40], demonstrating that this Snx3-retromer binding signal, which seemed to be an exception to the general rule, is not restricted to PC2 proteins [40]. Our basic phylogenetic analyses showed that an IEMxxΨ motif is present in Stx8 from all the *Schizosaccharomyces* species, and that the motif divergence coincides with the species divergence. Screening syntaxin 8 homologues for a loose Ψ+ΨxxΨ motif produced hits in a few fungal species but not in vertebrates. These data suggest that retromer sorting motifs might have undergone evolutionary divergence. The GxxIEMQxI motif was detected in PC2 from vertebrates, but not from *Drosophila* and *Caenorhabditis* species [40]. We have found loose sorting motifs related to GxxIEMQxI in polycystin-2 proteins from all the *Drosophila*, *Caenorhabditis* and *Schizosaccharomyces* species analyzed. In all cases, the closest species bear identical motifs, while distant species bear different motifs. Although these results point to evolutionary divergence, they are not conclusive. More specific analyses, and analyses that include other retromer sorting motifs will be required to determine whether the evolutionary divergence of the retromer-binding sequence is a general rule. Evolutionary divergence would explain why different retromer cargoes exhibit different sorting motifs, a situation that has not been observed in the sorting motifs recognized by the clathrin AP adaptors. Screening for a Ψ+ΨxxΨ motif might help to identify new retromer cargoes, and to reconsider the analysis of some cargoes that might have been disregarded because of the lack of potential motifs bearing aromatic residues. As mentioned above, the motifs in other homologues probably include a combination of large residues that would allow interaction with retromer and Snx3.

## Materials and methods

### Strains and growth conditions

All general growth conditions and yeast manipulations were performed as previously described [95,96]. The relevant genotypes and the source of the utilized strains are listed in the S1 Table. Unless otherwise stated, the experiments were performed with cells from cultures growing exponentially in liquid rich medium, yeast extract with supplements (YES; 0.5% yeast extract, 3% glucose, 225 mg/l adenine sulfate, histidine, leucine, uracil and lysine, and 2% agar). Geneticin (G418, Formedium), hygromycin (Formedium), and nourseothricin (Werner BioAgents) were used at 120 μg/ml, 400 μg/ml, and 50 μg/ml, respectively. Edimburg minimal medium with 20 mM glutamate (a good nitrogen source; EMMG) or 20 mM phenylalanine (a poor nitrogen source; MMPhe) instead of 93.5 mM $NH_4Cl$ (EMM2) were used to analyze cell

growth of prototrophic strains under nitrogen limiting conditions [49,50]. Latrunculin A (Sigma; stock at 5 mM in dimethyl sulfoxide [DMSO]) was used at 100 μM for 20 minutes.

## Genetic methods

Molecular and genetic manipulations were according to Sambrook et al. [97]. Gene deletions and tagged proteins were generated by transforming a *pku70Δ* strain [98] with polymerase chain reaction (PCR)-generated modules, as previously described [99]. The resulting transformants were backcrossed to reintroduce the *pku70⁺* allele. Different characters were combined by genetic crosses and selection of the traits of interest by random spore analysis [95]. The *fsv1⁺/stx8⁺* open reading frame (ORF) was PCR-amplified from genomic DNA with primers that introduced a *Mlu*I site upstream of the initial ATG and a *Sal*I site downstream of the Stop codon. This PCR fragment was fused in frame to the C-terminal end of Green Fluorescent Protein (GFP, amplified as an *Apa*I/*Mlu*I fragment) and cloned under the control of the α-tubulin *nda2⁺* 5′ untranslated region (UTR. A 571-base pair [bp] *Pst*I/*Apa*I DNA fragment) and 3′ UTR (a 313-bp *Sal*I/*Sac*I DNA fragment) into the pINTH1 vector [98], digested with *Pst*I and *Sac*I. The cassette was released by digestion with *Not*I and integrated into the exogenous *hph.171K* locus [98]. In this way, *stx8⁺* was constitutively expressed to allow a mild 4-fold overexpression (according to the information available in Pombase, https://www.pombase. org/; [68,70,100]). Several strategies were used to produce different *stx8* mutated versions. The N-terminal truncations shown in Fig 4D were generated by amplifying *stx8⁺* DNA fragments with primers that introduced *Mlu*I and *Sal*I sites at their 5′- and 3′-terminal ends, respectively, and cloned as described above. To produce the Stx8(Δ122–151) protein (Fig 5A), two DNA fragments with overlapping extensions corresponding to sequences 5′ upstream residue 122 and 3′ downstream residue 151 were generated (S1 File). These fragments were fused in a PCR reaction [101] using primers that introduced *Mlu*I and *Sal*I sites upstream of the ATG and downstream of the Stop codons, respectively, and cloned as described above. Fusion of extended PCR fragments was also used to change a series of 10 amino acids to alanine residues (Fig 5D) and to perform alanine-scanning mutagenesis (Fig 6A); in each case, the DNA extensions introduced the change of the corresponding amino acids to alanine residues. A similar approach was used to generate Stx8(Δ152–187) and Stx8(Δ188–224) deletions (Fig 7C). In all the cases, the final PCR product was flanked by *Mlu*I and *Sal*I sites and cloned as described for the *stx8⁺* allele. A DNA fragment corresponding to the Stx8(A+G) sequence (Fig 7E) flanked by *Mlu*I and *Sal*I sites (S1 File) was purchased as a custom gene from Integrated DNA Technologies (IDT, USA) and cloned as above. Technical reasons did not allow us to change all the residues of interest to alanine. Stx8(ΔSN) (Fig 7A) was generated by site-directed mutagenesis [102] using an oligo that flanked the sequences upstream of and downstream the SNARE domain. To avoid the defects in PVE morphology produced in *stx8* nonfunctional variants, which would make the observation of this organelle difficult, the localization of the GFP-Stx8 proteins described above was analyzed in a *stx8⁺* background. To produce an untagged Stx8 (ΔSN) protein from the *stx8⁺* locus under the control of its endogenous promoter, its coding sequence followed by the *nda2⁺* 3′ UTR was PCR-amplified from the pINTH+P*nda2⁺*:*stx8 (ΔSN)*:T*nda2⁺* plasmid and fused by PCR upstream of the *KANMX6* gene [99], which was in turn fused to a 600 bp DNA fragment corresponding to the sequence downstream of the *stx8⁺* 3'UTR (see S1 File). This cassette was used to transform yeast. To construct all Stx8 variants, DNA fragments were amplified using high fidelity Taq DNA polymerases following the manufacturers' recommendations. Velocity DNA polymerase (Bioline, UK) was used routinely, while PrimeSTAR HS DNA Polymerase (Takara Bio Inc, Japan) was used to perform fusion PCR. The accuracy of the constructions and integrations was assessed by DNA sequencing

and PCR, respectively. The sequence of each variant protein is shown in the S1 File. The expression level of all the GFP-Stx8 proteins was determined by western blotting (S7 Fig). For two-hybrid analyses ([103]. S7 File), genes were cloned as *Nde*I/*Sma*I DNA fragments into the pGADT7 and pGBKT7 plasmids (Clontech). *stx8* was amplified without the transmembrane helix; this variant was termed Stx8*. The *vps29-vps35-vps26* fusion gene (which was purchased as a custom gene from IDT, USA), was cloned as an *Nco*I/*Sma*I fragment into pGBKT7. AH109 *Saccharomyces cerevisiae* strain (Clontech) was used as a host. Transformants were selected on yeast nitrogen base (YNB. 0.7% yeast nitrogen base without amino acids, 2% glucose, 2% agar, pH 6.5) without leucine and tryptophan (YNB-L-T. 640 mg/L complete supplement mixture drop-out -LEU -TRP—Formedium, England—was added to the YNB medium). Protein-protein interaction was assessed as growth on YNB without histidine plates (YNB-H. 760 g/L complete supplement mixture drop-out -HIS was added to YNB). For bimolecular fluorescence (BiFC, [104]), the N-terminal half of Venus YFP was fused to the N-terminal end of Stx8 as explained above for GFP-Stx8, and the C-terminal half of Venus YFP was fused to the Snx3 C-terminal end as described ([99]. S8 File).

## Protein methods

Trichloroacetic acid (TCA) protein precipitation from cell extracts and western blot analysis were performed as described [41]. Cells growing exponentially in 30 ml YES were collected by centrifugation (900 x *g*), washed with 1 ml cold 20% TCA, and resuspended in 50 µl of the same solution. Five hundred µl of glass beads (Braun Biotech International) were added and the cells were broken in a cold Fast Prep FP120 (Savant Bio101) using three 16-second pulses (speed 6), with 5-minute incubations on ice between pulses. Four-hundred µl cold 5% TCA was added to the tube, which was vortexed to wash the beads. Cell extracts were transferred to a clean tube and centrifuged for 10 minutes at 4˚C. The pellets were resuspended in 2% sodium dodecyl sulfate (SDS)/0.3 M Tris base. Protein concentration was determined using Bradford protein assay reagent (Bio-Rad). Samples were equalized with respect to protein content and boiled in the presence of Laemmli sample buffer (50 mM Tris-HCl, pH 6.8; 1% SDS; 143 mM β-mercaptoethanol; and 10% glycerol) for 5 minutes. Samples were subjected to polyacrylamide gel electrophoresis (PAGE), transferred to polyvinylidene difluoride (PVDF) membranes, and incubated in blocking buffer (5% Nestlé non-fat dried milk in TBST: 0.25% Tris, pH 6.8; 0.9% NaCl; and 0.25% Tween 20) for 1 hour. Primary antibodies were anti-GFP (JL8, BD Living Colors; 1:3000) and anti-HA (12CA5, Roche; 1:5000). The secondary antibody was horseradish peroxidase-conjugated anti-mouse (Bio-Rad, 1:10000). Chemiluminescent signal was detected on X-ray films (Agfa) using the Western Bright ECL detection kit (Advansta). For coimmunoprecipitation, cells were washed with STOP buffer (10 mM ethylenediaminetetraacetic acid [EDTA], 154 mM NaCl, 10 mM NaF, and 10 mM NaN$_3$) and with washing buffer (20 mM 4-(2-hydroxyethyl)-1-piperazineethanesulfonic acid [HEPES], pH 7.5; 150 mM NaCl; and 5 mM EDTA), and broken in a 6770 Freezer/Mill (SPEX SamplePrep. Metuchen, USA) in lysis buffer (20 mM HEPES, pH 7.5; 150 mM NaCl; 5 mM EDTA; and 0.2% Triton X100). Cell debris were removed by centrifugation; 180 µg of the cleared cell lysates were boiled in the presence of Laemmli sample buffer and used as the "cell extract" samples. Three mg of lysates were treated with freshly made 1 mM dithiobis[succinimidyl propionate] (DSP) for 1 hour at 4˚C in a tube rotator and later incubated in the presence of 100 mM Tris-HCl at pH 7.5 for 15 minutes. Then, samples were immunoprecipitated with 50 µl anti-GFP µMAC magnetic beads (Miltenyi Biotec) following the manufacturer's recommendations, boiled in Laemmli sample buffer, and loaded into polyacrylamide gels as described above. For quantifications (Fig 5F), Fiji software (ImageJ. National Institutes of Health) was used to estimate the intensity of each

band. Then, the value for the Vps35-HA coimmunoprecipitates was divided by that of the GFP immunoprecipitates in the same sample. The final value for the Stx8(132–141)A variant was divided by that of Stx8.

## Microscopy

Staining with FM4-64 (Biotium) was performed as described [105]; cells were incubated in the presence of 10 µM of the dye for 1 hour at 30˚C. For conventional fluorescence microscopy, a Leica DM RXA microscope (63x objective; numerical aperture 1.4), equipped with a Photometrics Sensys CCD camera, was used; images were captured using Qfish 2.3 software. To obtain images with better resolution, an Olympus IX71 microscope (100× objective, numerical aperture 1.4) equipped with a personal DeltaVision system and a Photometrics CoolSnap HQ2 monochrome camera, was used; stacks of three Z-series sections corresponding to the cell middle were acquired at 0.2-µm intervals and images were processed using deconvolution Softworx DV software (Applied Precision). For confocal live-cell imaging, a spinning-disk Olympus IX-81 microscope equipped with a confocal CSUX1-A1 module (Yokogawa) and an Evolve (Photometrics) camera; images were acquired using Metamorph software. Tipically, to analyze protein colocalization, stacks of three 0.25 µm Z-sections of the cell middle were acquired, and the central plane of each stack was analyzed; the captured images were saved as 16-bit images, filtered with Fiji, and quantified using the JACoP plugin [106], adjusting the threshold for each channel and using an object-based method. To analyze the colocalization between GFP-Stx8 and Cfr1-RFP in the WT and *vps35Δ* strains (Fig 3E) avoiding the interference of vacuole signal, a threshold was manually adjusted for Cfr1-RFP pre-filtered images to establish the regions of interest (ROIs). Cfr1-RFP ROIs were overlapped to the pre-filtered GFP-Stx8 images, and the colocalization between Stx8 particles and Cfr1 ROIs was manually scored with the Fiji cell counter tool.

## Sequence analysis

*S. pombe* and *S. cerevisiae* sequences were obtained from Pombase (www.pombase.org/) and the *Saccharomyces* genome Database (SGD, www.yeastgenome.org/), respectively. Sequences from other organisms annotated as syntaxin 8 (Stx8) and PC2 (polycystin-2, Pkd2) were obtained from Uniprot (www.uniprot.org/) and the National Center for Biotechnology Information (www.ncbi.nlm.nih.gov/protein/). Multiple sequence alignments were performed with ClustalW at www.ebi.ac.uk/Tools/msa/clustalo/. Phylogram (midpoint rooted tree) of sequences aligned with clustal W were produced by using the PhyLM bootstrap option. Sequence logos were created using Weblogo 3 at weblogo.threeplusone.com/create.cgi using red font for acidic residues (E, D), blue for basic residues (H, K, R), green for bulky hydrophobic (aromatic) residues (W, Y, F), and purple for large and medium-volume hydrophobic residues (I, L, M, V). A custom motif search on selected proteins was performed with ScanProsite at prosite.expasy.org/scanprosite/ using Option 3 (Submit PROTEIN sequences and MOTIFS to scan them against each other).

## Supporting information

**S1 Fig. The levels of Stx8 and Vps35 are not altered in the *vps35Δ* and *stx8Δ* strains, respectively.** The asterisk denotes an unspecific band.
(TIF)

**S2 Fig. Ub:GFP-Cps1 is processed by vacuolar proteases.** Equal amounts of cell extracts from the indicated strains bearing the Ub:GFP-Cps1 construct were loaded into 8.5% PAGE

gels. The full-length protein and the cleaved GFP are denoted by arrows. The central and right panels are different exposures of the same blot.
(TIF)

**S3 Fig. Analysis of the functional relationship between Stx8 and Vsl1. (A)** Distribution of Vps10-GFP, GFP-Pep12 and Ub:GFP-Cps1 in WT and *vsl1Δ* strains. Images are medial sections acquired with a DeltaVision personal system. **(B)** Distribution of Vps10-GFP in WT and *snx8Δ* cells bearing an empty vector or a plasmid to overexpress *vsl1⁺*. Cells were grown in EMM without thiamine to allow overexpression. Images were captured with a Leica DM RXA conventional fluorescence microscope. Scale bar, 5 μm.
(TIF)

**S4 Fig. Two-hybrid analysis. (A)** Serial dilutions of the *S. cerevisiae* AH109 strain bearing plasmids that express two proteins known to interact—positive control (+)—, the two empty vectors (pGADT7 and pGBKT7)—negative control (-)—, controls expressing one vector and one of the proteins of interest, and plasmids bearing two of the proteins of interest (Stx8*, Vps26 and Snx3) were plated on YNB without leucine and tryptophan (YNB-L-T) and on YNB without histidine (YNB-H), and incubated at 30°C for four days. Lack of growth on YNB-H suggests no direct interaction or a very weak interaction. Stx8*, a Stx8 variant without the transmembrane helix. **(B)** Growth on YNB-L-T and YNB-H of the *S. cerevisiae* AH109 strain bearing empty vectors and/or the proteins of interest: Stx8*, Snx3 and CSC (Cargo-Selective Complex, a Vps29-Vps35-Vps26 fusion protein). Growth on YNB-H demonstrates direct interaction between Stx8* and the retromer CSC. It is noteworthy that the growth of the strain bearing Snx3 and the CSC complex is almost undetectable, even though direct Snx3-Vps26 interaction has been shown by Lucas et al [26].
(TIF)

**S5 Fig. Coimmunoprecipitation between Stx8 variants and Vps35.** For comparison, the interaction between Stx8 and Vps35 was included (lanes on the left). Cell extracts from *stx8Δ vps28Δ* strains carrying GFP-Stx8, GFP-Stx8(122–131), GFP-Stx8(132–141), GFP-Stx8(142–151) and/or Vps35-HA were analyzed by western blot using anti-GFP or anti-HA monoclonal antibodies before (WCE, whole-cell extracts) or after immunoprecipitation (IP) with a monoclonal anti-GFP antibody. The panels show the results of three independent experiments.
(TIF)

**S6 Fig. Complementation of *stx8Δ* sensitivity to KNO₃ by different clones of the indicated Stx8 variants.** Selected clones that show full complementation (Example 1) and partial complementation (Example 2) of the *stx8Δ* sensitivity to KNO₃ are shown. Plates were incubated at 32°C for three days.
(TIF)

**S7 Fig. Western blot of Stx8 variants.** Equal amounts of cell extracts from cells bearing the native Stx8 or different variants were subjected to western blotting using anti-GFP. Blotting with anti-tubulin was used as a loading control.
(TIF)

**S1 Table. List of strains used in this work.** Genotype and source of the strains used in this work.
(DOCX)

**S1 File. Construction of Stx8 variants.** Amino acidic sequence and information about the construction of the Stx8 variant
(DOCX)

**S2 File. Coiled-coils predictions.** NCOILS prediction of coiled-coils in Stx8 variants
(DOCX)

**S3 File. Sequence alignments used to generate the graphics shown in Fig 8.** The indicated sequence alignments were used to generate weblogos
(DOCX)

**S4 File. Syntaxin8 sequences used to screen for putative retromer-sorting motifs. The indicated sequences were used to screen for the presence of potential** Snx3-retromer sorting motifs using ScanProsite
(DOCX)

**S5 File. PC2 (Pkd2) sequences used to screen for putative retromer-sorting motifs. The indicated sequences were used to screen for the presence of potential** Snx3-retromer sorting motifs using ScanProsite.
(DOCX)

**S6 File. Conservation of the Vps26 and Snx3 residues relevant for interactions.** Sequence alignment of Vps26 and Snx3 sequences from *S. pombe*, *S. cerevisiae* and *H. sapiens*. The conserved residues are highlighted
(DOCX)

**S7 File. Sequence information related to the two-hybrid analysis.** Relevant sequence information about the two-hybrid analyses is provided.
(DOCX)

**S8 File. Sequence information related to the BiFC analysis.** Relevant sequence information about the BiFluorescence Complementation (BiFC) analyses is provided.
(DOCX)

## Acknowledgments

We thank L.L. Du, T. Kuno, I. Hagan, S. Labbé, P. Pérez, Y. Sanchez and the YGRC (http://yeast.lab.nig.ac.jp) for strains and plasmids, and S. Garcia-Dosil, M. Peña-Ramon, and A. Hernandez-Gonzalez for valuable technical help and some preliminary results.

## Author Contributions

**Conceptualization:** Francisco Yanguas, M.-Henar Valdivieso.

**Formal analysis:** Francisco Yanguas, M.-Henar Valdivieso.

**Funding acquisition:** M.-Henar Valdivieso.

**Investigation:** Francisco Yanguas, M.-Henar Valdivieso.

**Methodology:** Francisco Yanguas, M.-Henar Valdivieso.

**Project administration:** M.-Henar Valdivieso.

**Supervision:** M.-Henar Valdivieso.

**Writing – original draft:** M.-Henar Valdivieso.

**Writing – review & editing:** Francisco Yanguas, M.-Henar Valdivieso.

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
