## [Decision Letter · Decision Letter 0]

6 Oct 2020

Dear Dr Valdivieso,

Thank you very much for submitting your Research Article entitled 'Analysis of the SNARE Stx8 recycling reveals that the retromer-sorting motif has undergone evolutionary divergence' to PLOS Genetics. Your manuscript was fully evaluated at the editorial level and by independent peer reviewers. The reviewers appreciated the attention to an important problem, but raised some substantial concerns about the current manuscript. Based on the reviews, we will not be able to accept this version of the manuscript, but we would be willing to review again a much-revised version. We cannot, of course, promise publication at that time.

If you decide to revise the manuscript for further consideration at PLOS Genetics, please aim to resubmit within the next 60 days, unless it will take extra time to address the concerns of the reviewers, in which case we would appreciate an expected resubmission date by email to plosgenetics@plos.org.

[LINK]

We are sorry that we cannot be more positive about your manuscript at this stage. Please do not hesitate to contact us if you have any concerns or questions.

Yours sincerely,

Mara Duncan, Ph.D.

Guest Editor

PLOS Genetics

Gregory P. Copenhaver

Editor-in-Chief

PLOS Genetics

You manuscript has now been reviewed by two highly esteemed leaders in the field. Although reviewer #2 was enthusiastic about the broader importance of the diversity of retromer sorting motifs (and their evolution), both reviewers raised substantial concerns about the data supporting the main conclusions of the manuscript. In particular, a revised manuscript would need to demonstrate that Syt8 is a direct cargo of retromer a concern raised by both reviewers. In the absence of such evidence, this raises the possibility that a currently unknown linker protein, with a canonical retromer binding sequence, could link Syt8 to retromer as has been reported for other trafficking events such as AP-1 mediated traffic of E-cadherin via PIPKIγ acting as a linker (doi: 10.1083/jcb.200606023).

Equally important will be addressing Reviewer #2's concern that the study needs 'broader molecular phylogenetics analysis of retromer sorting motif(s) across several cargoes and other organisms' to support the paper's main conclusion regarding the divergence of sorting motifs.

In addition, Reviewer #2 suggests important experimental controls that should be addressed, including using a rescue construct (or segregant analysis) to verify that the phenotypes associated with Stx8Δ are caused by the deletion and not a carrier mutation, and verifying the degradation of Vps10 is due to lysosomal degradation.

Reviewer's Responses to Questions

**Comments to the Authors:**

Reviewer #1: In the study by Yanguas and Valdivieso a raft of data detailing the localization of the pombe Syntaxin 8 protein (Syt 8) is presented. The authors report that Syt 8 functions at the endosome and that the retromer complex mediates its trafficking between the endosome and the Golgi complex. They go on to show that a sorting motif in the Syt 8 protein is necessary for its localization and that the sorting motif is similar to one on the mammalian polycystin protein that has been reported to be a cargo of retromer.

I think the data, as presented, is largely solid and the conclusions are mostly well supported. I have two specific suggestions for experiments that, I feel, could strengthen this study:

1. The authors claim that Snx3-retromer is required for localizing Syt 8. This is not an unreasonable claim but I believe that the authors could enhance their study if they were to perform a native IP (as they have in figure 5 in the background of a Snx3delta (and possibly Vps5delta also). This would determine if it could be Snx3 making a direct contribution to cargo (i.e. syt 8) recognition as may be occurring. At present, it is not clear which of the retromer subunits is actually binding to Syt 8 (Vps35p? Vps26p? Vps5p? or perhaps Snx3p?) and investigating a little further which of the retromer proteins is responsible for binding to Syt 8 would be a worthwhile experiment.

2. Do the Syt 8 mutants that do not bind retromer (and therefore do not localize) possess any ability to rescue the salt-sensitivity phenotypes demonstrated by the Syt 8delta cells in figure 1? By examining if the mutants that can't bind retromer are salt-sensitive, the importance of the correct trafficking of Syt 8 to its function can be established and this should be a relatively simple experiment to carry out.

Reviewer #2: The cellular trafficking of transmembrane proteins is orchestrated through a variety of sorting motifs that interact with evolutionarily conserved protein complexes. Failure to interact with the correct sorting complex will lead to the miss-trafficking of the integral proteins (cargoes). The manuscript by Yanguas and Valdivieso explores the molecular evolution of a sorting motif for the retromer complex, a coat complex that contributes to the retrograde trafficking of cargoes from the endocytic system to the trans-Golgi network (TGN). The authors dissect the retromer sorting motif of the SNARE protein Stx8 in S. pombe and reveal that such a motif has undergone evolutionary divergence. The SNARE protein Stx8, homologue of the mammalian Syntaxin 8, functions in vesicle fusion at the pre-vacuolar endosome (PVE), this being the equivalent of late endosomes/lysosomes in mammalian cells. It is shown that Stx8 cycles between the PVE and the TGN, the yeast equivalent of early endosomes, and that this retrograde recycling is mediated by retromer subunits Vps35, Vps5, Vps17 and Snx3. The authors use truncation mutants and site directed mutants to identify the essential residues (133IEMeaM138) that compose the peptide sequence for the interaction with the retromer complex. Intriguingly, the retromer-binding sequence of Stx8 is distinct to the canonical retromer sorting motif Φx[L/M/V], but it shares similarities with that of Polycistin-2 (PC2), this being a retromer cargo with a unique GLEIEMQRI sorting signal. Based on the alignment of the two motifs the authors establish a common Gx[-]IEMxx(I/M) signal, and a degenerated (loose) Ψ[+/-]ΨxxΨP motif that could also accommodate the canonical retromer signal. Finally, bioinformatic analysis reveals the presence of such motif in few fungal Stx8 and showed that the divergence of these motifs matches the phylogenetic divergence of Schizosaccharomyces species.

Overall, the experiments appear technically sound, and the results are generally convincing (especially the identification and validation of the Stx8 sorting motif by use of truncation mutants and point mutants). In a few cases, additional controls may be needed to support specific conclusions, and some caveats may need to be stated. There are a number of issues with the paper that need to be clarified, most importantly the authors should avoid extending their conclusions beyond their own data, and avoid generalisations. For example, the identified Stx8 motif for retromer sorting has undergone evolutionary divergence, but this might not be the case for other retromer cargoes or other retromer-binding motifs.

The diversity of retromer sorting motifs (and their evolution) is an important aspect of the intracellular trafficking field that has long been underlooked. The findings in the present study are clearly of interest to understand the retrograde trafficking in yeast and, more broadly in eukaryotic cells, and thus provides important and relevant insight. We recommend publication if the caveats are addressed.

Major points:

The Ψ(E/D)ΨxxΨ motif is present in S. pombe Stx8 and H.sapiens PC2, but not in PC2 from D. melanogaster and C. elegans. The authors conclude that Ψ(E/D)ΨxxΨ has undergone evolutive divergence through metazoan and that the mammalian PC2 (almost-identical to that of Stx8) is a case of convergent evolution. How can the authors exclude that D. melanogaster and C. elegans have lost their strict Ψ(E/D)ΨxxΨ motif and are hence exceptions? The authors should comment on this.

The lack of a broader molecular phylogenetics analysis of retromer sorting motif(s) across several cargoes and other organisms makes it premature to draw conclusions regarding the evolutionary trend of the retromer motif(s). What the data of Yanguas and Valdivieso show is that the retromer motif is probably looser than what was thought before, and that some cargoes have more similar retromer sequences than others. We fully appreciate that the retromer-binding sequence of Stx8 has undergone evolutionary divergence, but a more systematic bioinformatic approach will be required to investigate whether this is an exception or whether this is the case for all other retromer-binding sequences. We suggest that the authors provide stronger evidence supporting the molecular evolution of the retromer motif or alternatively tone down their conclusions.

Figure 1 general. Have the protein levels of Stx8 and Vps35 (in the Stx8Δ and Vps35Δ) been validated with western blotting? Does the reduced intensity of Vps10-GFP signal in Stx8Δ correlate to an increased turnover/degradation of such protein in vacuoles. The authors should use quantitative western blotting to compare the total levels of Vps10-GFP in wt and Δ mutant lines.

Figure 1H. The results showed here are robust but the cleavage of free GFP could be an indirect effect (e.g. overexpression level). To make sure that the observation is indeed a vacuolar degradation phenotype, please rescue the phenotype with inhibitors of the vacuolar pathway (e.g. inhibitors of the vacuolar ATPase).

Figure 2. To confirm the specificity of the phenotype the authors should rescue the localisation of Vps10-GFP in the Stx8Δ cells by rescuing the phenotype with re-expression of Stx8 in the Δ cells.

Figure 3. Authors should confirm that the altered steady-state distribution of GFP-Stx8 in Vps35Δ cells, also correspond to a decreased co-localisation with a TGN marker.

Figure 5D and G. The biochemical experiments exploring the interaction between retromer and Stx8 should be solidified. We believe that it is important to repeat and quantify the degree of this interaction over at least n = 3 independent experiments, or alternatively strengthen the results by showing direct binding between Stx8 132-141 and the retromer complex.

Figure 8A. Please note that the sequence of PC2 is GLEIEMQRI instead of GIEIEMQRI. Also, the consensus between Stx8 and PC2 should be Gx[-]IEMxx(I/M) (both D and E are negatively charged). Please be consistent with the use of symbols +/- in the consensus sequences throughout the manuscript and figures.

Minor points:

It has been reported by Bean et al., 2017 that the retromer cargoes Ear1, Ymd8 and Ymr010w share a Ψ-E-(F/L) that could adhere to the first part of the Ψ[-]ΨxxΨ motif. Have the authors considered whether all these retromer binding sequences are just variation of a very loose and general retromer binding motif?

Have the authors considered testing some of the Vps26 mutants from Suzuki et al. 2019 to explore whether the retromer binding-sequence that they have identified in Stx8 binds the same surface of Vps26 that is involved in the interaction with Vps10 and/or Ear1?

The authors state that “it is possible that the IEMeaM motif and the TM domain have to be separated by a minimum number of amino acids”. This would be consistent with the recent structure by Kovtun et al., 2018 showing the core retromer subunits Vps26/35/29 are not directly in contact with the lipid membrane but layered on the top of Vps5 oligomers. Can the authors comment on the possibility that the linker between the TM domain and the retromer sorting motif is required so that the residues composing the sorting motif can reach the distal retromer subunits?

Bean BD, Davey M, Conibear E. Cargo selectivity of yeast sorting nexins. Traffic. 2017;18(2):110-122. doi:10.1111/tra.12459

Suzuki SW, Chuang YS, Li M, Seaman MNJ, Emr SD. A bipartite sorting signal ensures specificity of retromer complex in membrane protein recycling. J Cell Biol. 2019;218(9):2876-2886. doi:10.1083/jcb.201901019

Kovtun O, Leneva N, Bykov YS, et al. Structure of the membrane-assembled retromer coat determined by cryo-electron tomography. Nature. 2018;561(7724):561-564. doi:10.1038/s41586-018-0526-z

**Have all data underlying the figures and results presented in the manuscript been provided?**

Reviewer #1: Yes

Reviewer #2: Yes

PLOS authors have the option to publish the peer review history of their article (what does this mean?). If published, this will include your full peer review and any attached files.

Reviewer #1: No

Reviewer #2: No

---

## [Decision Letter · Decision Letter 1]

3 Mar 2021

Dear Dr Valdivieso,

We are pleased to inform you that your manuscript entitled "Analysis of the SNARE Stx8 recycling reveals that the retromer-sorting motif has undergone evolutionary divergence" has been editorially accepted for publication in PLOS Genetics. Congratulations!

Yours sincerely,

Mara Duncan, Ph.D.

Guest Editor

PLOS Genetics

Gregory P. Copenhaver

Editor-in-Chief

PLOS Genetics

Comments from the reviewers (if applicable):

Reviewer's Responses to Questions

**Comments to the Authors:**

Reviewer #1: I am satisfied with the revisions made by the authors.

Reviewer #2: The authors have responded in depth to our comments and have added substantial and important new data that addressed all our questions and concerns. The work by Yanguas and Valdivieso has been extensively expanded with new experimental data that further support their conclusions. A lot of effort was put into strengthening the biochemical evidence for a direct interaction between the Snx3-retormer complex and Stx8. Furthermore, a more thorough analysis of the phylogenesis of the retromer-binding motif has been carried out. By exploring the evolution of the retromer-sorting motif in several metazoans the authors have reinforced the idea of evolutionary divergence of the motif into each clade and between different groups. We thank the authors for their work and we recommend the publication of the manuscript.

**Have all data underlying the figures and results presented in the manuscript been provided?**

Reviewer #1: Yes

Reviewer #2: Yes

PLOS authors have the option to publish the peer review history of their article (what does this mean?). If published, this will include your full peer review and any attached files.

Reviewer #1: No

Reviewer #2: No

**Data Deposition**

http://datadryad.org/submit?journalID=pgenetics&manu=PGENETICS-D-20-01313R1

**Press Queries**

---

## [Editor Report · Acceptance letter]

26 Mar 2021

PGENETICS-D-20-01313R1 

Analysis of the SNARE Stx8 recycling reveals that the retromer-sorting motif has undergone evolutionary divergence 

Dear Dr Valdivieso, 

We are pleased to inform you that your manuscript entitled "Analysis of the SNARE Stx8 recycling reveals that the retromer-sorting motif has undergone evolutionary divergence" has been formally accepted for publication in PLOS Genetics! Your manuscript is now with our production department and you will be notified of the publication date in due course.

With kind regards,

Katalin Szabo

PLOS Genetics

On behalf of:
